# META-LEARNING WITH NEURAL TANGENT KERNELS

**Yufan Zhou**[*], **Zhenyi Wang**[*], **Jiayi Xian**, **Changyou Chen**, **Jinhui Xu** [†]
Department of Computer Science and Engineering, State University of New York at Buffalo
{yufanzho,zhenyiwa,jxian,changyou,jinhui}@buffalo.edu

## ABSTRACT

Model Agnostic Meta-Learning (MAML) has emerged as a standard framework for meta-learning, where a *meta-model* is learned with the ability of fast adapting to new tasks. However, as a double-looped optimization problem, MAML needs to differentiate through the whole inner-loop optimization path for every outer-loop training step, which may lead to both computational inefficiency and sub-optimal solutions. In this paper, we generalize MAML to allow meta-learning to be defined in function spaces, and propose the first meta-learning paradigm in the Reproducing Kernel Hilbert Space (RKHS) induced by the meta-model's Neural Tangent Kernel (NTK). Within this paradigm, we introduce two meta-learning algorithms in the RKHS, which no longer need a sub-optimal iterative inner-loop adaptation as in the MAML framework. We achieve this goal by 1) replacing the adaptation with a fast-adaptive regularizer in the RKHS; and 2) solving the adaptation analytically based on the NTK theory. Extensive experimental studies demonstrate advantages of our paradigm in both efficiency and quality of solutions compared to related meta-learning algorithms. Another interesting feature of our proposed methods is that they are demonstrated to be more robust to adversarial attacks and out-of-distribution adaptation than popular baselines, as demonstrated in our experiments.

## 1 INTRODUCTION

Meta-learning (Schmidhuber, 1987) has made tremendous progresses in the last few years. It aims to learn abstract knowledge from many related tasks so that fast adaption to new and unseen tasks becomes possible. For example, in few-shot learning, meta-learning corresponds to learning a meta-model or meta-parameters so that they can fast adapt to new tasks with a limited number of data samples. Among all existing meta-learning methods, Model Agnostic Meta-Learning (MAML) (Finn et al., 2017) is perhaps one of the most popular and flexible ones, with a number of follow-up works such as (Nichol et al., 2018; Finn et al., 2018; Yao et al., 2019; Khodak et al., 2019a;b; Denevi et al., 2019; Fallah et al., 2020; Lee et al., 2020; Tripuraneni et al., 2020). MAML adopts a double-looped optimization framework, where adaptation is achieved by one or several gradient-descent steps in the inner-loop optimization. Such a framework could lead to some undesirable issues related to computational inefficiency and sub-optimal solutions. The main reasons are that 1) it is computationally expensive to back-propagate through a stochastic-gradient-descent chain, and 2) it is hard to tune the number of adaptation steps in the inner-loop as it can be different for both training and testing. Several previous works tried to address these issues, but they can only alleviate them to certain extents. For example, first order MAML (FOMAML) (Finn et al., 2017) ignores the high-order terms of the standard MAML, which can speed up the training but may lead to deteriorated performance; MAML with Implicit Gradient (iMAML) (Rajeswaran et al., 2019) directly minimizes the objective of the outer-loop without performing the inner-loop optimization. But it still needs an iterative solver to estimate the meta-gradient.

To better address these issues, we propose two algorithms that generalize meta-learning to the Reproducing Kernel Hilbert Space (RKHS) induced by the meta-model's Neural Tangent Kernel (NTK) (Jacot et al., 2018). In this RKHS, instead of using parameter adaptation, we propose to perform an implicit function adaptation. To this end, we introduce two algorithms to avoid explicit

[*]The first two authors contribute equally. Correspondence to Changyou Chen (changyou@buffalo.edu).

[†]The research of the first and fifth authors was supported in part by NSF through grants CCF-1716400 and IIS-1910492.

function adaptation: one replaces the function adaptation step in the inner-loop with a new meta-objective with a fast-adaptive regularizer inspired by MAML; the other solves the adaptation problem analytically based on tools from NTK so that the meta-objective can be directly evaluated on samples in a closed-form. When restricting the function space to be RKHS, the solutions to the proposed two algorithms become conveniently solvable. In addition, we provide theoretical analysis on our proposed algorithms in the cases of using fully-connected neural networks and convolutional neural networks as the meta-model. Our analysis shows close connections between our methods and the existing ones. Particularly, we prove that one of our algorithms is closely related to MAML with some high-order terms ignored in the meta-objective function, thus endowing effective optimization. In summary, our main contributions are:

- We re-analyze the meta-learning problem and introduce two new algorithms for meta-learning in RKHS. Different from all existing meta-learning algorithms, our proposed methods can be solved efficiently without cumbersome chain-based adaptations.
- We conduct theoretically analysis on the proposed algorithms, which suggests that our proposed algorithms are closely related to the existing MAML methods when fully-connected neural networks and convolutional neural networks are used as the meta-model.
- We conduct extensive experiments to validate our algorithms. Experimental results indicate the effectiveness of our proposed methods, through standard few-shot learning, robustness to adversarial attacks and out-of-distribution adaptation.

## 2 PRELIMINARIES

### 2.1 META-LEARNING

Meta-learning can be roughly categorized as black-box adaptation methods (Andrychowicz et al., 2016; Graves et al., 2014; Mishra et al., 2018), optimization-based methods (Finn et al., 2017), non-parametric methods (Vinyals et al., 2016; Snell et al., 2017; Triantafillou et al., 2020) and Bayesian meta-learning methods (Finn et al., 2018; Yoon et al., 2018; Ravi & Beatson, 2019). In this paper, we focus on the framework of Model Agnostic Meta-Learning (MAML) (Finn et al., 2017), which has two key components, meta initialization and fast adaptation. Specifically, MAML solves the meta-learning problem through a double-looped optimization procedure. In the inner-loop, MAML runs a task-specific adaptation procedure to transform a meta-parameter, $\boldsymbol{\theta}$, to a task-specific parameter, $\{\boldsymbol{\phi}_m\}_{m=1}^B$, for a total of $B$ different tasks. In the outer-loop, MAML minimizes a total loss of $\sum_{m=1}^B \mathcal{L}(f_{\boldsymbol{\phi}_m})$ with respect to meta-parameter $\boldsymbol{\theta}$, where $f_{\boldsymbol{\phi}_m}$ is the model adapted on task $m$ that is typically represented by a deep neural network. It is worth noting that in MAML, one potential problem is to compute the meta-gradient $\nabla_{\boldsymbol{\theta}} \sum_{m=1}^B \mathcal{L}(f_{\boldsymbol{\phi}_m})$. It requires one to differentiate through the whole inner-loop optimization path, which could be very inefficient.

### 2.2 GRADIENT FLOW

Our proposed method relies on the concept of gradient flow. Generally speaking, gradient flow is a continuous-time version of gradient descent. In the finite-dimensional parameter space, a gradient flow is defined by an ordinary differential equation (ODE), $d\boldsymbol{\theta}^t/dt = -\nabla_{\boldsymbol{\theta}^t} F(\boldsymbol{\theta}^t)$, with a starting point $\boldsymbol{\theta}^0$ and function $F : R^d \to R$. Gradient flow is also known as steepest descent curve.

One can generalize gradient flows to infinite-dimensional function spaces. Specifically, given a function space $\mathcal{H}$, a functional $\mathcal{F} : \mathcal{H} \to R$, and a starting point $f^0 \in \mathcal{H}$, a gradient flow is similarly defined as the solution of $df^t/dt = -\nabla_{f^t} \mathcal{F}(f^t)$. This is a curve in the function space $\mathcal{H}$. In this paper, we use notation $\nabla_{f^t} \mathcal{F}(f^t)$, instead of $\nabla_{\mathcal{H}} \mathcal{F}(f^t)$, to denote the general function derivative of the energy functional $\mathcal{F}$ with respect to function $f^t$ (Villani, 2008).

### 2.3 THE NEURAL TANGENT KERNEL

Neural Tangent Kernel (NTK) is a recently proposed technique for characterizing the dynamics of a neural network under gradient descent (Jacot et al., 2018; Arora et al., 2019; Lee et al., 2019). NTK allows one to analyze deep neural networks (DNNs) in RKHS induced by NTK. One immediate benefit of this is that the loss functional in the function space is often convex, even when it is highly non-convex in the parameter space (Jacot et al., 2018) [*]. This property allows one to better understand the property of DNNs. Specifically, let $f_{\boldsymbol{\theta}}$ be a DNN parameterized by $\boldsymbol{\theta}$. The corresponding NTK $\boldsymbol{\Theta}$

---

[*]Let $\mathcal{H}$ be the function space, $F$ be the realization function for neural network defined in Section 3.2. Note even if a functional loss (*e.g.*, L2 loss) $\mathcal{E} : \mathcal{H} \to R$ is convex on $\mathcal{H}$, the composition $\mathcal{E} \circ F$ is in general not.

is defined as: $\boldsymbol{\Theta}(\mathbf{x}_1, \mathbf{x}_2) = \dfrac{\partial f_{\boldsymbol{\theta}}(\mathbf{x}_1)}{\partial \boldsymbol{\theta}} \dfrac{\partial f_{\boldsymbol{\theta}}(\mathbf{x}_2)}{\partial \boldsymbol{\theta}}^{\mathsf{T}}$, where $\mathbf{x}_1, \mathbf{x}_2$ are two data points. In our paper, we will define meta-learning on an RKHS induced by such a kernel.

## 3 META-LEARNING IN RKHS

We first define the meta-learning problem in a general function space, and then restrict the function space to be an RKHS, where two frameworks will be proposed to make meta-learning feasible in RKHS, along with some theoretical analysis. For simplicity, in the following we will hide the superscript time $t$ unless necessary, *e.g.*, when the analysis involves time-changing.

### 3.1 META-LEARNING IN FUNCTION SPACE

Given a function space $\mathcal{H}$, a distribution of tasks $P(\mathcal{T})$, and a loss function $\mathcal{L}$, the goal of meta-learning is to find a *meta function* $f^* \in \mathcal{H}$, so that it performs well after simple adaptation on a specific task. Let $\mathcal{D}_m^{tr}$ and $\mathcal{D}_m^{test}$ be the training and testing sets, respectively, sampled from a data distribution of task $\mathcal{T}_m$. The meta-learning problem on function space $\mathcal{H}$ is defined as:

$$f^* = \underset{f \in \mathcal{H}}{\arg\min}\, \mathcal{E}(f), \text{ with } \mathcal{E}(f) = \mathbb{E}_{\mathcal{T}_m}\left[ \mathcal{L}\Big(\mathsf{Adapt}(f, \mathcal{D}_m^{tr}), \mathcal{D}_m^{test}\Big) \right] \tag{1}$$

where $\mathsf{Adapt}$ denotes some adaptation algorithms, *e.g.*, several steps of gradient descent; $\mathcal{E} : \mathcal{H} \to R$ is called energy functional, which is used to evaluate the model represented by the function $f$.

In theory, solving equation 1 is equivalent to solving the gradient flow equation $\mathrm{d}f^t/\mathrm{d}t = -\nabla_{f^t}\mathcal{E}(f^t)$. However, solving the gradient flow equation is generally infeasible, since *i*) it is hard to directly apply optimization methods in function space and *ii*) the energy functional $\mathcal{E}$ contains an adaptation algorithm $\mathsf{Adapt}$, making the functional gradient infeasible. Thus, a better way is to design a special energy functional so that it can be directly optimized without running the specific adaptation algorithm. In the following, we first specify the functional meta-learning problem in RKHS, and then propose two methods to derive efficient solutions for the problem.

### 3.2 META-LEARNING IN RKHS

We consider a function $f$ that is parameterized by $\boldsymbol{\theta} \in \mathbb{R}^P$, denoted as $f_{\boldsymbol{\theta}}$, with $P$ being the number of parameters. Define a realization function $F : \mathbb{R}^P \to \mathcal{H}$ that maps parameters to a function. With these, we can then define an energy function in the parameter space as $E \triangleq \mathcal{E} \circ F : R^P \to R$ with $\circ$ being the composition operator. Consequently, with an initialized $\boldsymbol{\theta}^0$, we can define the gradient flow of $E(\boldsymbol{\theta}^t)$ in parameter space as: $\mathrm{d}\boldsymbol{\theta}^t/\mathrm{d}t = -\nabla_{\boldsymbol{\theta}^t}E(\boldsymbol{\theta}^t)$. In the following, we first establish an equivalence between the gradient flow in RKHS and the gradient flow in the parameter space. We then propose two algorithms for meta-learning in the RKHS induced by NTK.

**Theorem 1** *Let $\mathcal{H}$ be an RKHS induced by the NTK $\boldsymbol{\Theta}$ of $f_{\boldsymbol{\theta}}$. With $f^0 = f_{\boldsymbol{\theta}^0}$, the gradient flow of $\mathcal{E}(f^t)$ coincides with the function evolution of $f_{\boldsymbol{\theta}^t}$ driven by the gradient flow of $E(\boldsymbol{\theta}^t)$.*

The proof of Theorem 1 relies on the property of NTK (Jacot et al., 2018), and is provided in the Appendix. Theorem 1 serves as a foundation of our proposed methods, which indicates that solving the meta-learning problem in RKHS can be done by some appropriate manipulations. In the following, we describe two different approaches termed Meta-RKHS-I and Meta-RKHS-II, respectively.

### 3.3 META-RKHS-I: META-LEARNING IN RKHS WITHOUT ADAPTATION

Our goal is to design an energy functional that has no adaptation component, but is capable of achieving fast adaptation. For this purpose, we first introduce two definitions: empirical loss function $\mathcal{L}(f_{\boldsymbol{\theta}}, \mathcal{D}_m)$ and expected loss function $\mathcal{L}(f_{\boldsymbol{\theta}})$. Let $\mathcal{D}_m = \{\mathbf{x}_{m,i}, \mathbf{y}_{m,i}\}_{i=1}^n$ be a set containing the data of a regression task $\mathcal{T}_m$. The empirical loss function $\mathcal{L}(f_{\boldsymbol{\theta}}, \mathcal{D}_m)$ and the expected loss function $\mathcal{L}_m(f_{\boldsymbol{\theta}})$ can be defined as:

$$\mathcal{L}(f_{\boldsymbol{\theta}}, \mathcal{D}_m) = \frac{1}{2n}\sum_{i=1}^n \left\| f(\mathbf{x}_{m,i}) - \mathbf{y}_{m,i} \right\|^2, \quad \mathcal{L}_m(f_{\boldsymbol{\theta}}) = \mathbb{E}_{\mathbf{x}_m, \mathbf{y}_m}\left[ \frac{1}{2}\left\| f(\mathbf{x}_m) - \mathbf{y}_m \right\|^2 \right].$$

Our idea is to define a regularized functional such that it endows the ability of fast adaptation in RKHS. Our solution is based on some property of the standard MAML. We start from analyzing the meta-objective of MAML with a $k$-step gradient-descent adaptation, *i.e.*, applying $k$ gradient-descent steps in the inner-loop. The objective can be formulated as

$$\boldsymbol{\theta}^* = \arg\min_{\boldsymbol{\theta}} \mathbb{E}_{\mathcal{T}_m}\left[\mathcal{L}(f_{\boldsymbol{\phi}}, \mathcal{D}_m^{test})\right], \text{ with } \boldsymbol{\phi} = \boldsymbol{\theta} - \alpha\sum_{i=0}^{k-1}\nabla_{\boldsymbol{\theta}_i}\mathcal{L}(f_{\boldsymbol{\theta}_i}, \mathcal{D}_m^{tr}),$$

where $\alpha$ is the learning rate of the inner-loop, $\boldsymbol{\theta}_0 = \boldsymbol{\theta}$, and $\boldsymbol{\theta}_{i+1} = \boldsymbol{\theta}_i - \alpha\nabla_{\boldsymbol{\theta}_i}\mathcal{L}(f_{\boldsymbol{\theta}_i}, \mathcal{D}_m^{tr})^\dagger$. By Taylor expansion, we have

$$\mathbb{E}_{\mathcal{T}_m}\left[\mathcal{L}(f_{\boldsymbol{\phi}}, \mathcal{D}_m^{test})\right] \approx \mathbb{E}_{\mathcal{T}_m}\left[\mathcal{L}(f_{\boldsymbol{\theta}}, \mathcal{D}_m^{test}) - \alpha\sum_{i=0}^{k-1}\nabla_{\boldsymbol{\theta}_i}\mathcal{L}(f_{\boldsymbol{\theta}_i}, \mathcal{D}_m^{tr})\nabla_{\boldsymbol{\theta}}\mathcal{L}(f_{\boldsymbol{\theta}}, \mathcal{D}_m^{test})^\mathsf{T}\right]. \quad (2)$$

Since $\mathcal{D}_m^{tr}$ and $\mathcal{D}_m^{test}$ come from the same distribution, equation 2 is an unbiased estimator of

$$\mathcal{M}_k = \mathbb{E}_{\mathcal{T}_m}[\mathcal{L}_m(f_{\boldsymbol{\theta}}) - \sum_{i=0}^{k-1}\beta_i], \text{ where } \beta_i = \alpha\nabla_{\boldsymbol{\theta}_i}\mathcal{L}_m(f_{\boldsymbol{\theta}_i})\nabla_{\boldsymbol{\theta}}\mathcal{L}_m(f_{\boldsymbol{\theta}})^\mathsf{T}. \quad (3)$$

We focus on the case of $k = 1$, which is $\mathcal{M}_1 = \mathbb{E}_{\mathcal{T}_m}[\mathcal{L}_m(f_{\boldsymbol{\theta}})] - \alpha\mathbb{E}_{\mathcal{T}_m}[\|\nabla_{\boldsymbol{\theta}}\mathcal{L}_m(f_{\boldsymbol{\theta}})\|^2]$. The first term on the RHS is the traditional multi-task loss evaluated at $\boldsymbol{\theta}$ for all tasks. The second term corresponds to the negative gradient norm; minimizing it means choosing a $\boldsymbol{\theta}$ with the maximum gradient norm. Intuitively, when $\boldsymbol{\theta}$ is not a stationary point of a task, one should choose the steepest descent direction to reduce the loss maximally for a specific task, thus leading to *fast adaptation*.

The above understanding suggests us to propose the following regularized energy functional, $\widetilde{\mathcal{E}}_\alpha$, for meta-learning in the RKHS induced with the NTK for *fast function adaptation*:

$$\widetilde{\mathcal{E}}(\alpha, f_{\boldsymbol{\theta}}) = \mathbb{E}_{\mathcal{T}_m}\left[\mathcal{L}_m(f_{\boldsymbol{\theta}}) - \alpha\|\nabla_{f_{\boldsymbol{\theta}}}\mathcal{L}_m(f_{\boldsymbol{\theta}})\|_{\mathcal{H}}^2\right], \quad (4)$$

where $\|\cdot\|_{\mathcal{H}}$ denotes the functional norm in $\mathcal{H}$, and $\alpha$ is a hyper-parameter. The above objective is inspired by the Taylor expansion of the MAML objective, but is defined in the RKHS induced by the NTK. Its connection with MAML and some functional-space properties will be discussed later.

**Solving the Function Optimization Problem**    To minimize equation 4, we first derive Theorem 2 to reduce the function optimization problem to a parameter optimization problem.

**Theorem 2** *Let $f_{\boldsymbol{\theta}}$ be a neural network with parameter $\boldsymbol{\theta}$ and $\mathcal{H}$ be the RKHS induced by the NTK $\Theta$ of $f_{\boldsymbol{\theta}}$. Then, the following are equivalent*

$$\widetilde{\mathcal{E}}(\alpha, f_{\boldsymbol{\theta}}) = \mathcal{M}_1, \text{ and } \|\nabla_{f_{\boldsymbol{\theta}}}\mathcal{L}_m(f_{\boldsymbol{\theta}})\|_{\mathcal{H}}^2 = \|\nabla_{\boldsymbol{\theta}}\mathcal{L}_m(f_{\boldsymbol{\theta}})\|^2.$$

Theorem 2 is crucial to our approach as it indicates that solving problem equation 4 is no more difficult than the original parameter-based MAML, although it only considers one-step adaptation case. Next, we will show that multi-step adaptation in the parameter space can also be well-approximated by our objective equation 4 but with a scaled regularized parameter $\alpha$. In the following, we consider the squared loss $\mathcal{L}$. The case with the cross-entropy loss is discussed in the Appendix. We assume that $f_{\boldsymbol{\theta}}$ is parameterized by either fully-connected or convolutional neural networks, and only consider the impact of number of hidden layers $L$ in our theoretical results.

**Theorem 3** *Let $f_{\boldsymbol{\theta}}$ be a fully-connected neural network with $L$ hidden layers and ReLU activation function, $s_1, ..., s_{L+1}$ be the spectral norm of the weight matrices, $s = \max_h s_h$, and $\alpha$ be the learning rate of gradient descent. If $\alpha \leq O(qr)$ with $q = \min(1/(Ls^L), L^{-1/(L+1)})$ and $r = \min(s^{-L}, s)$, then the following holds*
$$|\mathcal{M}_k - \widetilde{\mathcal{E}}(k\alpha, f_{\boldsymbol{\theta}})| \leq O\left(\frac{1}{L}\right).$$

**Theorem 4** *Let $f_{\boldsymbol{\theta}}$ be a convolutional neural network with $L - l$ convolutional layers and $l$ fully-connected layers and with ReLU activation function, and $d_x$ be the input dimension. Denote by $W^h$ the parameter **vector** of the convolutional layer for $h \leq L - l$, and the weight **matrices** of the fully connected layers for $L - l + 1 \leq h \leq L + 1$. $\|\cdot\|_2$ means both the spectral norm of a matrix*

---

$^\dagger$For ease of our later notation, we write the gradient $\nabla_{\boldsymbol{\theta}_i}\mathcal{L}$ (thus the parameter as well) as a row vector.

and the Euclidean norm of a vector. Define $s_h = \sqrt{d_x}\|W^h\|_2$ if $h = 1, ..., L - l$, and $\|W^h\|_2$ if $L - l + 1 \leq h \leq L + 1$. Let $s = \max_h s_h$ and $\alpha$ be the learning rate of gradient descent. If $\alpha \leq O(qr)$ with $q = \min(1/(Ls^L), L^{-1/(L+1)})$ and $r = \min(s^{-L}, s)$, the following holds

$$|\mathcal{M}_k - \widetilde{\mathcal{E}}(k\alpha, f_{\boldsymbol{\theta}})| \leq O\Big(\frac{1}{L}\Big).$$

The above Theorems indicate that, for a meta-model with fully-connected and convolutional layers, the proposed Meta-RKHS-I can be an efficient approximation of MAML with a bounded error.

**Comparisons with Reptile and MAML** Similar to Reptile and MAML, the testing stage of Meta-RKHS-I also requires gradient-based adaptation on meta-test tasks. By Theorem 1, we known that gradient flow of an energy functional can be approximated by gradient descent in a parameter space. Reptile with 1-step adaptation (Nichol et al., 2018) is equivalent to the approximation of the gradient flow of $\widetilde{\mathcal{E}}(\alpha, f_{\boldsymbol{\theta}})$ with $\alpha = 0$, which does not include the fast-adaptation regularization as in our method. For a fairer comparison on the efficiency, we will discuss the computational complexity later.

From the equivalent parameter-optimization form indicated in Theorem 2, we know that our energy functional $\widetilde{\mathcal{E}}(\alpha, f_{\boldsymbol{\theta}})$ is closely related to MAML. However, with this form, our method does not need the explicit adaptation steps in training (*i.e.*, the inner-loop of MAML), leading to a simpler optimization problem. We will show that our proposed method leads to better results.

### 3.4 META-RKHS-II: META-LEARNING IN RKHS WITH A CLOSED-FORM ADAPTATION

In this section, we present our second solution for meta-learning in RKHS by deriving a closed-form adaptation function, *i.e.*, we focus on a case where $\mathsf{Adapt}(f, \mathcal{D}_m^{tr})$ is analytically solvable using the theory of NTK. Specifically, we are given a loss function $\mathcal{L}$, tasks $\mathcal{T}_m$ with randomly split training set $\mathcal{D}_m^{tr} = \{\mathbf{x}_{m,i}^{tr}, \mathbf{y}_{m,i}^{tr}\}_{i=1}^n$, and testing set $\mathcal{D}_m^{test}$. Let $\boldsymbol{\theta}_m^t$ and $f_{m,\boldsymbol{\theta}}^t$ denote the parameters and the corresponding function at time $t$ adapted by task $\mathcal{T}_m$ from the meta parameter $\boldsymbol{\theta}$ and meta function $f_{\boldsymbol{\theta}}$, respectively. From the NTK theory (Jacot et al., 2018; Arora et al., 2019; Lee et al., 2019), we can write the function/parameter evolution as:

$$\frac{\mathrm{d}\boldsymbol{\theta}_m^t}{\mathrm{d}t} = -\nabla_{\boldsymbol{\theta}_m^t} \mathcal{L}(f_{m,\boldsymbol{\theta}}^t, \mathcal{D}_m^{tr}), \quad \text{and} \quad \frac{\mathrm{d}f_{m,\boldsymbol{\theta}}^t}{\mathrm{d}t} = \frac{\mathrm{d}\boldsymbol{\theta}_m^t}{\mathrm{d}t}\frac{\partial f_{m,\boldsymbol{\theta}}^t}{\partial \boldsymbol{\theta}_m^t}^{\mathsf{T}} = \sum_{i=1}^n \frac{\partial \mathcal{L}(f_{m,\boldsymbol{\theta}}^t, \mathcal{D}_m^{tr})}{\partial f_{m,\boldsymbol{\theta}}^t(\mathbf{x}_{m,i}^{tr})} \Theta(\mathbf{x}_{m,i}^{tr}, \cdot).$$

The above differential equation corresponds to the adaptation step, *i.e.*, how to adapt the meta parameter/function for task $m$. By the NTK theory, we can show that this admits closed-form solutions. In our meta-learning settings, this indicates that no explicit adaptation steps are necessary.

To see why this is the case, we first investigate the regression case, where the loss function $\mathcal{L}$ is the squared loss. Let $\mathbf{x} \in \mathcal{D}_m^{test}$ be a test data point. As shown in Arora et al. (2019); Lee et al. (2019), with a large enough neural network we can safely assume that NTK will not change too much during the training. In this case, we can have a closed-form solution for $f_{m,\boldsymbol{\theta}}^t$ as

$$f_{m,\boldsymbol{\theta}}^t(\mathbf{x}) = f_{\boldsymbol{\theta}}(\mathbf{x}) + H(\mathbf{x}, \mathbf{X}_m^{tr})H^{-1}(\mathbf{X}_m^{tr}, \mathbf{X}_m^{tr})\left(e^{-tH(\mathbf{X}_m^{tr}, \mathbf{X}_m^{tr})} - \mathbf{I}\right)\left(f_{\boldsymbol{\theta}}(\mathbf{X}_m^{tr}) - Y^{tr}\right), \quad (5)$$

where $e$ is the matrix exponential map, which can be approximated by $Pad\acute{e}$ approximation (M.Arioli et al., 1996). $H(\mathbf{X}_m^{tr}, \mathbf{X}_m^{tr})$ is an $n \times n$ kernel matrix with its $(i, j)$ element being $\Theta(\mathbf{x}_{m,i}, \mathbf{x}_{m,j})$, $H(\mathbf{x}, \mathbf{X}_m^{tr})$ is a $1 \times n$ vector with its $i$-th element being $\Theta(\mathbf{x}, \mathbf{x}_{m,i})$, $f_{\boldsymbol{\theta}}(\mathbf{X}_m^{tr}) \in R^n$ is the predictions of all training data at the initialization, and $Y^{tr} \in R^n$ is the target value of the training data. Specifically, at time $t = \infty$, we have

$$f_{m,\boldsymbol{\theta}}^{\infty}(\mathbf{x}) = f_{\boldsymbol{\theta}}(\mathbf{x}) + H(x, \mathbf{X}_m^{tr})H^{-1}(\mathbf{X}_m^{tr}, \mathbf{X}_m^{tr})\left(Y^{tr} - f_{\boldsymbol{\theta}}(\mathbf{X}_m^{tr})\right). \quad (6)$$

The above results allow us to directly define an energy functional by substituting $\mathsf{Adapt}(f, \mathcal{D}_m^{tr})$ in equation 1 with its closed-form solution $f_{m,\boldsymbol{\theta}}^t$. In other words, our new energy functional is

$$\overline{\mathcal{E}}(t, f_{\boldsymbol{\theta}}) = \mathbb{E}_{\mathcal{T}_m}\left[\mathcal{L}_m(f_{m,\boldsymbol{\theta}}^t)\right], \quad (7)$$

where $f_{m,\boldsymbol{\theta}}^t$ is defined in equation 5, and $\mathcal{L}_m(f_{m,\boldsymbol{\theta}}^t)$ is the expectation of $\mathcal{L}\left(f_{m,\boldsymbol{\theta}}^t, \mathcal{D}_m^{test}\right)$. For classification problems, we follow the same strategy as in Arora et al. (2019) to extend regression to classification. Mores details can be found in the Appendix, including the algorithm in Appendix A.

Table 1: Running time comparison per iteration with $C_1 = d_x p + L p^2$ and $C_2 = d_x p + L d_x p^2$.

|  | FOMAML | Reptile | Meta-RKHS-I | Meta-RKHS-II |
|---|---|---|---|---|
| Fully-connected | $O(n(k+1)C_1)$ | $O(nkC_1)$ | $O(nC_1)$ | $O(nC_1 + n^3)$ |
| Convolutional | $O(n(k+1)C_2)$ | $O(nkC_2)$ | $O(nC_2)$ | $O(nC_2 + n^3)$ |

**On Potential Robustness of Meta-RKHS-II**  Our extensive empirical studies show that Meta-RKHS-II is a more robust model than related baselines. We provide an intuitive explanation on the potential robustness of Meta-RKHS-II, as we find current theories of both robustness machine learning and NTK are insufficient for a formal explanation. Our explanation is based on some properties of both the meta-learning framework and NTK: 1) Strong initialization (meta model): For NTK to generalize well, we argue that it is necessary to start the model with a good initialization. This is automatically achieved in our meta-learning setting, where the meta model serves as the initialization for NTK predictions. Actually, this has been supported by recent research (Fort et al., 2020), which shows that there is a chaotic stage in the NTK prediction with finite neural networks, and the NTK regime can be reachable with a good initialization. 2) Low complex classification boundary: It is known that NTK is a linear model in the NTK regime. Intuitively, generating adversarial samples with a lower complex model should be relatively harder because there is less data in the vicinity of the decision boundary compared to a more complex model, making the probability of the model being attacked smaller. Thus we argue that our model can be more robust than standard meta learning models. 3) Our NTK-based model is robust enough to adapt with different time steps. And these finite time steps can be more robust to adversarial attacks than that of the infinite-time limit partly due to the complexity of back-propagating gradients. We note each of the individual factors might not be enough to ensure robustness. Instead, we argue it is the combination effect of these factors that lead to robustness of our model. Formal analysis is out of the scope of this paper and left for future work.

**Connection with Meta-RKHS-I**  The proposed two methods choose different strategies to avoid explicit adaptation in meta-learning, which seem to be two very different algorithms. We prove below theorem, which indicates that the difference of the underlying gradient flows of the two algorithms indeed increases w.r.t. both $T$ and the depth $L$ of a DNN (we only consider impacts of $T$ and $L$).

**Theorem 5** *Let $f_{\boldsymbol{\theta}}$ be a neural network with $L$ hidden layers, with each layer being either fully-connected or convolutional. Assume that $\|\mathcal{L}\|_\infty < \infty$. Then, $error(T) = |\widetilde{\mathcal{E}}(T, f_{\boldsymbol{\theta}}) - \overline{\mathcal{E}}(T, f_{\boldsymbol{\theta}})|$ is a non-decreasing function of $T$. Furthermore, for arbitrary $T > 0$ we have $error(T) \leq O\big(T^{2L+3}\big)$.*

Actually, Meta-RKHS-II implicitly contains a term of functional gradient norm because $\overline{\mathcal{E}}(T, f_{\boldsymbol{\theta}}) = \mathbb{E}_{\mathcal{T}_m} \left[ \mathcal{L}_m(f_{\boldsymbol{\theta}}) - \int_0^T \left\| \nabla_{\boldsymbol{\theta}^t} \mathcal{L}_m(f_{m,\boldsymbol{\theta}}^t) \right\|^2 \mathrm{d}t \right]$. The difference compared to Meta-RKHS-I mainly comes from the fact that Meta-RKHS-I can be regarded as an approximation of time-discrete adaptation, while Meta-RKHS-II is based on time-continuous adaptation. In our experiments, we observe that Meta-RKHS-I is as fast as FOMAML, which means that it is more computationally efficient than the standard MAML. Meanwhile Meta-RKHS-II is the more robust model in tasks of adversarial attack and out-of-distribution adaptation.

**Connection with iMAML**  Our proposed method is similar to the iMAML algorithm (Finn & Levine, 2019) in the sense that both methods try to solve meta-learning without executing the optimization path. Different from iMAML, which still relies on an iterative solver, our method only needs to solve a simpler optimization problem due to the closed-form adaptation.

## 3.5 TIME COMPLEXITY ANALYSIS

We compare the time complexity of our proposed methods with other first-order meta-learning methods. Without loss of generality, we analyze the complexity in the case of a $L$-layer MLP or $L$-layer convolutional neural networks. Recall that $d_x$ is the input dimension. Assume each layer has width (filter number) $O(p)$. Let $n$ be the data batch size, $k$ the adaptation steps of inner-loop optimization. We summarize the time complexity in Table 1, where we simply assume the complexity of multiplying matrices with sizes $a \times b$ and $b \times c$ to be $O(abc)$. Note in the meta-learning setting, $n$ is typically small, indicating the efficiency of our proposed methods.

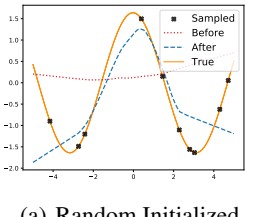 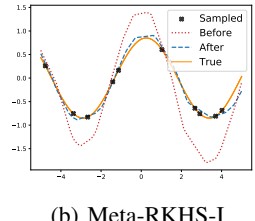 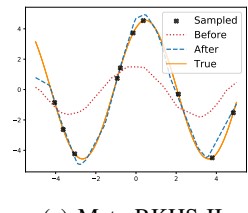

| (a) Random Initialized | (b) Meta-RKHS-I | (c) Meta-RKHS-II |

Figure 1: Performance of random initialized network and our methods. The models before/after adaptation are shown in dotted/dashed lines, samples used for adaptation are also shown in the figure.

## 4 EXPERIMENTS

We conduct a set of experiments to evaluate the effectiveness of our proposed methods, including a sine wave regression toy experiment, few-shot classification, robustness to adversarial attacks, out-of-distribution generalization and ablation study. Due to space limit, more results are provided in the Appendix. We compare our models with related baselines including MAML (Finn et al., 2017), the first order MAML (FOMAML) (Finn et al., 2017), Reptile (Nichol et al., 2018) and iMAML (Rajeswaran et al., 2019). Results are reported as mean and variance over three independent runs.

### 4.1 REGRESSION

Following Finn et al. (2017); Nichol et al. (2018), we first test our proposed methods on the 1-dimensional sine wave regression problem. This problem is instructive, where a model is trained on many different sine waves with different amplitudes and phases, and tested by adapting the trained model to new sine waves with only a few data points using a fixed number of gradient-descent steps. Following Finn et al. (2017); Nichol et al. (2018), we use a fully-connected neural network with 2 hidden layers and the ReLU activation function. The results are shown in Figure 1.

### 4.2 FEW-SHOT IMAGE CLASSIFICATION

For this experiment, we choose two popular datasets adopted for meta-learning: Mini-ImageNet and FC-100 (Oreshkin et al., 2018). The cross-entropy loss is adopted for Meta-RKHS-I; while the squared loss is used for Meta-RKHS-II following Arora et al. (2019); Novak et al. (2019). Similar to Finn et al. (2017), the model architecture is set to be a four-layer convolutional neural network with ReLU activation. The filter number is set to be 32. The Adam optimizer (Kingma & Ba, 2015) is used to minimize the energy functional. Meta batch size is set to be 16 and learning rates are set to be 0.01 for Meta-RKHS-II.

The results are shown in Table 2. Note the results of Reptile is different from those in Nichol et al. (2018), because we re-evaluate it under the same setting as Finn et al. (2017), *i.e.*, 10 steps of adaptation is applied during testing. Our results of iMAML is based on the implementation of Spigler (2019). It is observed that our proposed methods

Table 2: Few-shot classification results on Mini-ImageNet and FC-100.

| | MINI-IMAGENET | | FC-100 | |
| ALGORITHM | 5 WAY 1 SHOT | 5 WAY 5 SHOTS | 5 WAY 1 SHOT | 5 WAY 5 SHOTS |
| --- | --- | --- | --- | --- |
| MAML | $48.70 \pm 1.84\%$ | $63.11 \pm 0.93\%$ | $38.00 \pm 1.95\%$ | $49.34 \pm 0.97\%$ |
| FOMAML | $48.07 \pm 1.75\%$ | $63.15 \pm 0.91\%$ | $37.73 \pm 1.93\%$ | $49.05 \pm 0.99\%$ |
| IMAML | $49.30 \pm 1.88\%$ | $64.89 \pm 0.95\%$ | $38.38 \pm 1.70\%$ | $49.41 \pm 0.80\%$ |
| REPTILE | $49.70 \pm 1.83\%$ | $65.91 \pm 0.84\%$ | $38.40 \pm 1.94\%$ | $50.50 \pm 0.87\%$ |
| META-RKHS-I | $\mathbf{51.10 \pm 1.82}\%$ | $\mathbf{66.19 \pm 0.80}\%$ | $38.90 \pm 1.90\%$ | $\mathbf{51.47 \pm 0.86}\%$ |
| META-RKHS-II | $50.53 \pm 2.09\%$ | $65.40 \pm 0.91\%$ | $\mathbf{41.20 \pm 2.17}\%$ | $51.36 \pm 0.96$ |

achieve better accuracy than different baselines. Interestingly, our Meta-RKHS-I performs better than FOMAML (this is also the case in other experiments), although they share a similar objective. We conjecture the reason is because our Meta-RKHS-I restricts the function to be in an RKHS, making the functional space smaller thus easier to optimize compared to the unrestricted version of FOMAML. In terms of our two algorithms, there is not always a winner on all the tasks. We note that Meta-RKHS-I is more efficient in training. However, we show below that Meta-RKHS-II is better in terms of robustness to adversarial attacks and out-of-distribution generalization.

### 4.3 ROBUSTNESS TO ADVERSARIAL ATTACKS

We now compare the adversarial robustness of our methods with other popular baselines. We adopt both white-box and black-box attacks in this experiment. For the white-box attacks, we adopt strong attacks including the PGD Attack (Madry et al., 2017), BPDA attack (Athalye et al., 2018) and

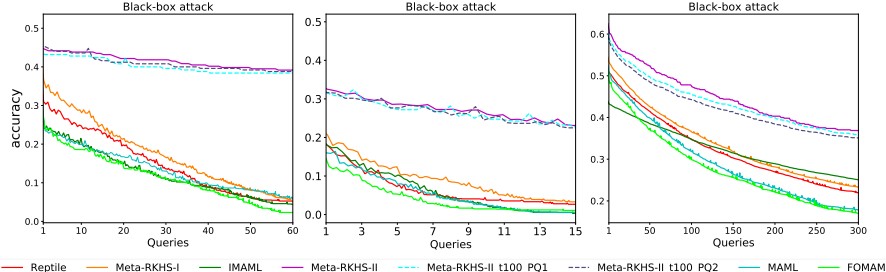

Figure 2: Black-box attack on Mini-ImageNet and FC-100. Mini-ImageNet 5-way 1-shot (left), FC-100 5-way 1-shot (middle) and Mini-ImageNet 5-way 5-shot (right).

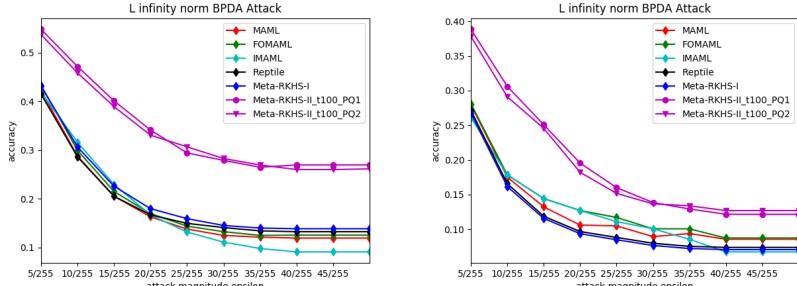

Figure 3: BPDA attack on Mini-ImageNet 5-way 5-shot (left) and FC-100 5-way 5-shot (right).

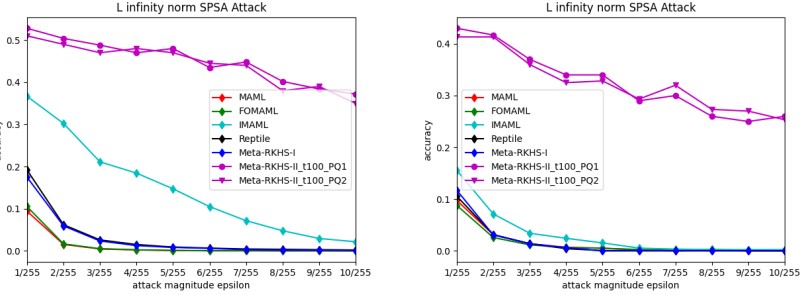

Figure 4: SPSA attack on Mini-ImageNet 5-way 5-shot (left) and FC-100 5-way 5-shot (right).

SPSA attack (Uesato et al., 2018). For PGD attack, we use $\ell_\infty$ norm and compare the results on Mini-imagenet and FC-100. We compare the robust accuracy with different magnitude with 20-step attack with a step size of $2/255$. For BPDA attack, we apply median smoothing, JPEGFilter and BitSqueezing as input transformation adapted from (Guo et al., 2018) as defense strategies. For SPSA attack, we follow (Uesato et al., 2018) and set the Adam learning rate 0.01, perturbation size $\delta = 0.01$. For Black-box attack, we adopt the strong query efficient attack method (Guo et al., 2019). Follow the setting of Guo et al. (2019), we use a fixed step size of 0.2.

We consider both finite-time and infinite-time adaptation in this experiment. For finite-time adaptation, the Padé approximation with $P = Q = 1$ and $P = Q = 2$ to approximate the matrix exponential are considered (Butcher & Chipman, 1992). We use Meta-RKHS-II_t100_PQ1 and Meta-RKHS-II_t100_PQ2 to denote methods using finite time $t = 100$, $P = Q = 1$ or $P = Q = 2$, respectively. We observe other finite time $t$ makes similar predictions, thus we only consider $t = 100$. The results from the black-box attack in Figure 2 indicate the robustness of our Meta-RKHS-II. In fact, the gaps are significantly large, making it the only useful robust model in the adversarial-attack setting. Our Meta-RKHS-I is not as robust as Meta-RKHS-II, but still slightly outperforms other baselines. Regarding the white-box attack, results in Figure 3, 4 and 5 again show that our proposed Meta-RKHS-II is significantly more robust than baselines under the three strong attacks. It is also interesting to see that our Meta-RKHS-I performs slightly better than Meta-RKHS-II in some rare cases, *e.g.*, in the Mini-ImageNet 5-way 1-shot case when the attack magnitude is not too small. More results are presented in the Appendix.

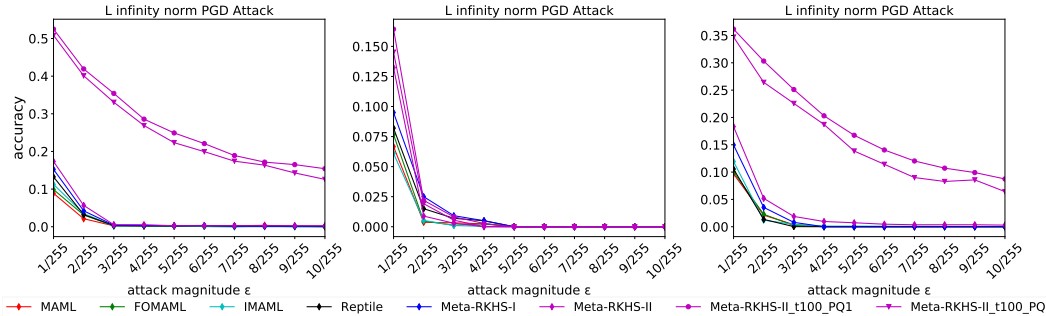

Figure 5: $\ell_\infty$ norm PGD attack on Mini-ImageNet and FC-100. Mini-ImageNet 5-way 5-shot (left), Mini-ImageNet 5-way 1-shot (middle) and FC-100 5-way 5-shot (right).

Table 4: Meta-RKHS-II with different time $t$.

|  | TIME $t$ | $t = 0.1$ | $t = 1$ | $t = 10$ | $t = 100$ | $t = \infty$ |
|---|---|---|---|---|---|---|
| MINI-IMAGENET | 5 WAY 1 SHOT | $49.67 \pm 2.23\%$ | $48.27 \pm 2.23\%$ | $\mathbf{50.53 \pm 2.09}\%$ | $49.13 \pm 2.19\%$ | $48.70 \pm 2.28\%$ |
|  | 5 WAY 5 SHOTS | $64.51 \pm 0.93\%$ | $64.28 \pm 0.98\%$ | $\mathbf{65.40 \pm 0.91}\%$ | $64.24 \pm 1.06\%$ | $64.95 \pm 0.96\%$ |
| FC-100 | 5 WAY 1 SHOT | $36.50 \pm 2.10\%$ | $38.80 \pm 2.32\%$ | $\mathbf{41.20 \pm 2.17}\%$ | $38.80 \pm 2.21\%$ | $37.60 \pm 2.13\%$ |
|  | 5 WAY 5 SHOTS | $48.35 \pm 1.02\%$ | $49.79 \pm 1.04\%$ | $\mathbf{51.36 \pm 0.96}\%$ | $48.59 \pm 1.09\%$ | $49.48 \pm 0.98\%$ |

## 4.4 OUT-OF-DISTRIBUTION GENERALIZATION

We adopt similar strategy in (Lee et al., 2020) to test a model's ability of generalizing to out-of-distribution datasets. In this setting, the state of arts are achieved by Bayesian TAML (Lee et al., 2020). Different from their setting that considers any-shot learning with maximum number of examples for each class being as large as 50, we only focus on the standard 1 or 5 shot learning. We thus modify their code to accommodate our standard setting. The CUB (Wah et al., 2011) and VGG Flower Nilsback & Zisserman (2008) are fine-grained datasets used in this experiment, where all images are resized to $84 \times 84$. We follow Lee et al. (2020) to split these datasets into meta training/validation/testing sets. We first train all the methods on Mini-ImageNet or FC-100 datasets, then conduct meta-testing on CUB and VGG Flower datasets. The results are shown in Table 3. Again, our methods achieve the best results, outperforming the state-of-art method with our Meta-RKHS-II, indicating the robustness of our proposed methods. More results are presented in the Appendix.

## 4.5 ABLATION STUDY

We conduct several ablation studies, including: comparing Reptile with Meta-RKHS-I under different adaptation steps (results shown in the Appendix), testing the impact of choosing different time $t$ in Meta-RKHS-II (results shown in Table 4) and the impact of network architecture with dif-

Table 3: Meta testing on different out-of-distribution datasets with model trained on Mini-ImageNet.

| ALGORITHM | 5 WAY 1 SHOT | | 5 WAY 5 SHOT | |
|---|---|---|---|---|
|  | CUB | VGG FLOWER | CUB | VGG FLOWER |
| MAML | $34.23 \pm 1.52\%$ | $52.98 \pm 1.76\%$ | $52.36 \pm 0.94\%$ | $67.52 \pm 1.30\%$ |
| FOMAML | $35.32 \pm 1.69\%$ | $53.86 \pm 1.64\%$ | $52.02 \pm 0.71\%$ | $68.83 \pm 1.16\%$ |
| REPTILE | $35.61 \pm 1.38\%$ | $53.57 \pm 1.58\%$ | $51.93 \pm 0.89\%$ | $71.62 \pm 1.25\%$ |
| IMAML | $40.55 \pm 0.61\%$ | $54.97 \pm 0.80\%$ | $46.31 \pm 2.03\%$ | $60.67 \pm 1.91\%$ |
| BAYESIAN TAML(SOTA) | $41.57 \pm 0.60\%$ | $58.56 \pm 0.66\%$ | $61.78 \pm 0.56\%$ | $77.95 \pm 0.46\%$ |
| META-RKHS-I | $36.73 \pm 1.26\%$ | $54.79 \pm 1.61\%$ | $54.19 \pm 0.73\%$ | $72.76 \pm 1.08\%$ |
| META-RKHS-II | $\mathbf{45.36 \pm 0.87}\%$ | $\mathbf{60.80 \pm 1.02}\%$ | $\mathbf{65.21 \pm 0.64}\%$ | $\mathbf{78.25 \pm 0.49}\%$ |

ferent number of CNN feature channels (results shown in the Appendix). It is interesting to see that a finite-time (around $t = 10$) achieves the best accuracy, although the infinite-time case guarantees a stationary point. This indicates that a stationary point achieved by limited training data in the adaptation step is not always the best choice, because the limited training data might easily overfit the model, thus achieving worse test results.

## 5 CONCLUSION

We develop meta-learning in RKHS, and propose two practical algorithms allowing efficient adaptation in the function space by avoiding some complicated adaptations as in traditional methods. We show connections between our proposed methods and existing ones. Extensive experiments suggest that our methods are more effective, achieve better generalization and are more robust against adversarial attacks and out-of-distribution generalization, compared to popular strong baselines.

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

## A  ALGORITHMS

Our proposed algorithms for meta-learning in the RKHS are summarized in Algorithm 1.

---
**Algorithm 1** Meta-Learning in RKHS

---
**Require:** $p(\mathcal{T})$: distribution over tasks, randomly initialized neural network parameters $\boldsymbol{\theta}$.
  **while** not done **do**
    Sample a batch of tasks $\{\mathcal{T}_m\}_{m=1}^B \sim p(\mathcal{T})$
    **for all** $\mathcal{T}_m$ **do**
      Sample a batch of data points $\mathcal{D}_m$ **or** Sample two batches of data points $\mathcal{D}_m^{tr}, \mathcal{D}_m^{test}$.
    **end for**
    Evaluate the energy functional by equation 4 with $\{\mathcal{D}_m\}_{m=1}^B$ **or** Evaluate the energy functional
    by equation 7 with $\{\mathcal{D}_m^{tr}, \mathcal{D}_m^{test}\}_{m=1}^B$. Minimize the energy functional w.r.t $\boldsymbol{\theta}$.
  **end while**

---

## B  PROOF OF THEOREM 1

**Theorem 1** *If $f_{\boldsymbol{\theta}}$ is a neural network with parameter $\boldsymbol{\theta} \in R^P$ and $\mathcal{H}$ is the Reproducing Kernel Hilbert Space (RKHS) induced by $\boldsymbol{\Theta}$, where $\boldsymbol{\Theta}$ is the Neural Tangent Kernel (NTK) of $f_{\boldsymbol{\theta}}$, then with initialization $f^0 = f_{\boldsymbol{\theta}^0}$, the gradient flow of $\mathcal{E}(f^t)$ coincides with the function evolution of $f_{\boldsymbol{\theta}^t}$ induced by the gradient flow of $E(\boldsymbol{\theta}^t)$.*

**Proof** Without loss of generality, we can rewrite $\mathcal{E}(f) = \mathbb{E}_{\mathcal{T}_m}\{\mathbb{E}_{(\mathbf{x}_m, \mathbf{y}_m)}[C(f(\mathbf{x}_m), \mathbf{y}_m)]\}$ with some function $C(\cdot, \cdot)$.

For a neural network $f_{\boldsymbol{\theta}}$ with parameter $\boldsymbol{\theta} \in R^P$, the gradient flow of $E$ in $R^P$ is

$$\frac{\mathrm{d}\boldsymbol{\theta}^t}{\mathrm{d}t} = -\nabla_{\boldsymbol{\theta}^t} E(\boldsymbol{\theta}^t).$$

We have

$$
\begin{aligned}
\frac{\mathrm{d}\boldsymbol{\theta}^t}{\mathrm{d}t} &= -\nabla_{\boldsymbol{\theta}^t}(\mathcal{E} \circ F)(\boldsymbol{\theta}^t) \\
&= -\mathbb{E}_{\mathcal{T}_m}\{\mathbb{E}_{(\mathbf{x}_m, \mathbf{y}_m)}[\nabla_{\boldsymbol{\theta}^t} C(f_{\boldsymbol{\theta}^t}(\mathbf{x}_m), \mathbf{y}_m)]\} \\
&= -\mathbb{E}_{\mathcal{T}_m}\left\{\mathbb{E}_{(\mathbf{x}_m, \mathbf{y}_m)}\left[\frac{\partial C(f_{\boldsymbol{\theta}}^t(\mathbf{x}_m), \mathbf{y}_m)}{\partial f_{\boldsymbol{\theta}}^t(\mathbf{x}_m)}\frac{\partial f_{\boldsymbol{\theta}}^t(\mathbf{x}_m)}{\partial \boldsymbol{\theta}^t}\right]\right\}.
\end{aligned}
$$

We know that the dynamics of $f_{\boldsymbol{\theta}^t}$ is

$$
\begin{aligned}
\frac{\mathrm{d}f_{\boldsymbol{\theta}^t}}{\mathrm{d}t} &= \frac{\mathrm{d}\boldsymbol{\theta}^t}{\mathrm{d}t}\frac{\partial f_{\boldsymbol{\theta}^t}}{\partial \boldsymbol{\theta}^t}^{\mathsf{T}} \\
&= -\mathbb{E}_{\mathcal{T}_m}\left\{\mathbb{E}_{(\mathbf{x}_m, \mathbf{y}_m)}\left[\frac{\partial C(f_{\boldsymbol{\theta}^t}(\mathbf{x}_m), \mathbf{y}_m)}{\partial f_{\boldsymbol{\theta}^t}(\mathbf{x}_m)}\frac{\partial f_{\boldsymbol{\theta}^t}(\mathbf{x}_m)}{\partial \boldsymbol{\theta}^t}\right]\right\}\frac{\partial f_{\boldsymbol{\theta}^t}}{\partial \boldsymbol{\theta}^t}^{\mathsf{T}} \\
&= -\mathbb{E}_{\mathcal{T}_m}\left\{\mathbb{E}_{(\mathbf{x}_m, \mathbf{y}_m)}\left[\frac{\partial C(f_{\boldsymbol{\theta}^t}(\mathbf{x}_m), \mathbf{y}_m)}{\partial f_{\boldsymbol{\theta}^t}(\mathbf{x}_m)}\frac{\partial f_{\boldsymbol{\theta}^t}(\mathbf{x})}{\partial \boldsymbol{\theta}^t}\frac{\partial f_{\boldsymbol{\theta}^t}}{\partial \boldsymbol{\theta}^t}^{\mathsf{T}}\right]\right\} \\
&= -\mathbb{E}_{\mathcal{T}_m}\left\{\mathbb{E}_{(\mathbf{x}_m, \mathbf{y}_m)}\left[\frac{\partial C(f_{\boldsymbol{\theta}^t}(\mathbf{x}_m), \mathbf{y}_m)}{\partial f_{\boldsymbol{\theta}^t}(\mathbf{x}_m)}\boldsymbol{\Theta}^t(\mathbf{x}_m, \cdot)\right]\right\},
\end{aligned}
\tag{8}
$$

where $\boldsymbol{\Theta}^t$ is the Neural Tangent Kernel of neural network $f_{\boldsymbol{\theta}^t}$ (Jacot et al., 2018).

If $\mathcal{H}^t$ is the Reproducing Kernel Hilbert Space induced by a kernel $\boldsymbol{\Theta}^t$ and $V_{\mathbf{x}_m} : \mathcal{H} \to R$ is the evaluation functional at $\mathbf{x}_m$, which is defined as

$$V_{\mathbf{x}_m}(f) = f(\mathbf{x}_m),$$

then for an arbitrary function $g$ and a small perturbation $\epsilon$, we have

$$\langle \nabla_f V_{\mathbf{x}_m}(f), g \rangle = \lim_{\epsilon \to 0} \frac{V_{\mathbf{x}_m}(f + \epsilon g) - V_{\mathbf{x}_m}(f)}{\epsilon}$$

$$\langle \nabla_f V_{\mathbf{x}_m}(f), g \rangle = \lim_{\epsilon \to 0} \frac{f(\mathbf{x}_m) + \epsilon g(\mathbf{x}_m) - f(\mathbf{x}_m)}{\epsilon}$$

$$\langle \nabla_f V_{\mathbf{x}_m}(f), g \rangle = g(\mathbf{x}_m)$$

$$\langle \nabla_f V_{\mathbf{x}_m}(f), g \rangle = \langle \mathbf{\Theta}^t(\mathbf{x}_m, \cdot), g \rangle$$

$$\nabla_f V_{\mathbf{x}_m}(f) = \mathbf{\Theta}^t(\mathbf{x}_m, \cdot)$$

$$\nabla_f f(\mathbf{x}_m) = \mathbf{\Theta}^t(\mathbf{x}_m, \cdot).$$

With an initial function $f^0 = f_{\boldsymbol{\theta}^0} \in \mathcal{H}$, the gradient flow of $\mathcal{E}$ in $\mathcal{H}$ is

$$\frac{\mathrm{d}f^t}{\mathrm{d}t} = -\nabla_{f^t} \mathcal{E}(f^t).$$

We have

$$\frac{\mathrm{d}f^t}{\mathrm{d}t} = -\mathbb{E}_{\mathcal{T}_m} \{ \mathbb{E}_{(\mathbf{x}_m, \mathbf{y}_m)} \left[ \nabla_{f^t} C(f^t(\mathbf{x}_m), \mathbf{y}_m) \right] \}$$

$$= -\mathbb{E}_{\mathcal{T}_m} \left\{ \mathbb{E}_{(\mathbf{x}_m, \mathbf{y}_m)} \left[ \frac{\partial C(f^t(\mathbf{x}_m), \mathbf{y}_m)}{\partial f^t(\mathbf{x}_m)} \nabla_{f^t} f^t(\mathbf{x}_m) \right] \right\}$$

$$= -\mathbb{E}_{\mathcal{T}_m} \left\{ \mathbb{E}_{(\mathbf{x}_m, \mathbf{y}_m)} \left[ \frac{\partial C(f^t(\mathbf{x}_m), \mathbf{y}_m)}{\partial f^t(\mathbf{x}_m)} \mathbf{\Theta}^t(\mathbf{x}_m, \cdot) \right] \right\}. \tag{9}$$

We can complete the proof by comparing equation 8 and equation 9. ∎

## C    PROOF OF THEOREM 2

**Theorem 2** *If $f_{\boldsymbol{\theta}}$ is a neural network with parameter $\boldsymbol{\theta}$ and $\mathcal{H}$ is the Reproducing Kernel Hilbert Space (RKHS) induced by $\mathbf{\Theta}$, where $\mathbf{\Theta}$ is the Neural Tangent Kernel (NTK) of $f_{\boldsymbol{\theta}}$, then*

$$\mathcal{M}_1 = \widetilde{\mathcal{E}}(\alpha, f_{\boldsymbol{\theta}}), \text{ and } \beta_0 = \alpha \|\nabla_{\boldsymbol{\theta}} \mathcal{L}_m(f_{\boldsymbol{\theta}})\|^2 = \alpha \|\nabla_{f_{\boldsymbol{\theta}}} \mathcal{L}_m(f_{\boldsymbol{\theta}})\|_{\mathcal{H}}^2.$$

**Proof**  Without loss of generality, we rewrite $\mathcal{L}_m(f_{\boldsymbol{\theta}}) = \mathbb{E}_{\mathbf{x}_m, \mathbf{y}_m} [C(f_{\boldsymbol{\theta}}(\mathbf{x}_m), \mathbf{y}_m)]$.

In regression task, we have

$$C(f_{\boldsymbol{\theta}}(\mathbf{x}_m), \mathbf{y}_m) = \frac{1}{2} \|f_{\boldsymbol{\theta}}(\mathbf{x}_m) - \mathbf{y}_m\|^2$$

. In classification task, we have

$$C(f_{\boldsymbol{\theta}}(\mathbf{x}_m), \mathbf{y}_m) = \mathbf{y}_m \log(f_{\boldsymbol{\theta}}(\mathbf{x}_m))^{\mathsf{T}},$$

where log is element-wise logarithm operation.

$$\|\nabla_{\boldsymbol{\theta}} \mathcal{L}_m(f_{\boldsymbol{\theta}})\|^2$$

$$= \nabla_{\boldsymbol{\theta}} \mathcal{L}_m(f_{\boldsymbol{\theta}}) \nabla_{\boldsymbol{\theta}} \mathcal{L}_m(f_{\boldsymbol{\theta}})^{\mathsf{T}}$$

$$= \nabla_{\boldsymbol{\theta}} \mathbb{E}_{\mathbf{x}_m, \mathbf{y}_m} [C(f_{\boldsymbol{\theta}}(\mathbf{x}_m), \mathbf{y}_m)] \nabla_{\boldsymbol{\theta}} \mathbb{E}_{\mathbf{x}_m, \mathbf{y}_m} [C(f_{\boldsymbol{\theta}}(\mathbf{x}_m), \mathbf{y}_m)]^{\mathsf{T}}$$

$$= \mathbb{E}_{\mathbf{x}_m, \mathbf{y}_m} \left[ \frac{\partial C(f_{\boldsymbol{\theta}}(\mathbf{x}_m), \mathbf{y}_m)}{\partial f_{\boldsymbol{\theta}}(\mathbf{x}_m)} \frac{\partial f_{\boldsymbol{\theta}}(\mathbf{x}_m)}{\partial \boldsymbol{\theta}} \right] \mathbb{E}_{\mathbf{x}_m, \mathbf{y}_m} \left[ \frac{\partial f_{\boldsymbol{\theta}}(\mathbf{x}_m)}{\partial \boldsymbol{\theta}}^{\mathsf{T}} \frac{\partial C(f_{\boldsymbol{\theta}}(\mathbf{x}_m), \mathbf{y}_m)}{\partial f_{\boldsymbol{\theta}}(\mathbf{x}_m)}^{\mathsf{T}} \right]$$

$$= \mathbb{E}_{\mathbf{x}_m, \mathbf{y}_m} \left\{ \mathbb{E}_{\mathbf{x}'_m, \mathbf{y}'_m} \left[ \frac{\partial C(f_{\boldsymbol{\theta}}(\mathbf{x}_m), \mathbf{y}_m)}{\partial f_{\boldsymbol{\theta}}(\mathbf{x}_m)} \frac{\partial f_{\boldsymbol{\theta}}(\mathbf{x}_m)}{\partial \boldsymbol{\theta}} \frac{\partial f_{\boldsymbol{\theta}}(\mathbf{x}'_m)}{\partial \boldsymbol{\theta}}^{\mathsf{T}} \frac{\partial C(f_{\boldsymbol{\theta}}(\mathbf{x}'_m), \mathbf{y}'_m)}{\partial f_{\boldsymbol{\theta}}(\mathbf{x}'_m)}^{\mathsf{T}} \right] \right\}$$

$$= \mathbb{E}_{\mathbf{x}_m, \mathbf{y}_m} \left\{ \mathbb{E}_{\mathbf{x}'_m, \mathbf{y}'_m} \left[ \frac{\partial C(f_{\boldsymbol{\theta}}(\mathbf{x}_m), \mathbf{y}_m)}{\partial f_{\boldsymbol{\theta}}(\mathbf{x}_m)} \mathbf{\Theta}(\mathbf{x}_m, \mathbf{x}'_m) \frac{\partial C(f_{\boldsymbol{\theta}}(\mathbf{x}'_m), \mathbf{y}'_m)}{\partial f_{\boldsymbol{\theta}}(\mathbf{x}'_m)}^{\mathsf{T}} \right] \right\}$$

$$= \left\langle \mathbb{E}_{\mathbf{x}_m, \mathbf{y}_m} \left[ \frac{\partial C(f_{\boldsymbol{\theta}}(\mathbf{x}_m), \mathbf{y}_m)}{\partial f_{\boldsymbol{\theta}}(\mathbf{x}_m)} \boldsymbol{\Theta}(\mathbf{x}_m, \cdot) \right], \mathbb{E}_{\mathbf{x}'_m, \mathbf{y}'_m} \left[ \frac{\partial C(f_{\boldsymbol{\theta}}(\mathbf{x}'_m), \mathbf{y}'_m)}{\partial f_{\boldsymbol{\theta}}(\mathbf{x}'_m)} \boldsymbol{\Theta}(\mathbf{x}'_m, \cdot) \right] \right\rangle_{\mathcal{H}}$$

$$= \left\langle \mathbb{E}_{\mathbf{x}_m, \mathbf{y}_m} \left[ \frac{\partial C(f_{\boldsymbol{\theta}}(\mathbf{x}_m), \mathbf{y}_m)}{\partial f_{\boldsymbol{\theta}}(\mathbf{x}_m)} \nabla_{f_{\boldsymbol{\theta}}} f_{\boldsymbol{\theta}}(\mathbf{x}_m) \right], \mathbb{E}_{\mathbf{x}'_m, \mathbf{y}'_m} \left[ \frac{\partial C(f_{\boldsymbol{\theta}}(\mathbf{x}'_m), \mathbf{y}'_m)}{\partial f_{\boldsymbol{\theta}}(\mathbf{x}'_m)} \nabla_{f_{\boldsymbol{\theta}}} f_{\boldsymbol{\theta}}(\mathbf{x}'_m) \right] \right\rangle_{\mathcal{H}}$$

$$= \left\langle \nabla_{f_{\boldsymbol{\theta}}} \mathcal{L}_m(f_{\boldsymbol{\theta}}), \nabla_{f_{\boldsymbol{\theta}}} \mathcal{L}_m(f_{\boldsymbol{\theta}}) \right\rangle_{\mathcal{H}}$$

$$= \| \nabla_{f_{\boldsymbol{\theta}}} \mathcal{L}_m(f_{\boldsymbol{\theta}}) \|_{\mathcal{H}}^2,$$

where $\langle \cdot, \cdot \rangle_{\mathcal{H}}$ is the inner product in Reproducing Kernel Hilbert Space (RKHS) $\mathcal{H}$. In the above equations, we use the definition of Neural Tangent Kernel (NTK), the property of inner product in RKHS, the definition of evaluation functional and its gradient in RKHS.

Recall that

$$\widetilde{\mathcal{E}}(\alpha, f_{\boldsymbol{\theta}}) = \mathbb{E}_{\mathcal{T}_m} \left[ \mathcal{L}_m(f_{\boldsymbol{\theta}}) - \alpha \| \nabla_{f_{\boldsymbol{\theta}}} \mathcal{L}_m(f_{\boldsymbol{\theta}}) \|_{\mathcal{H}}^2 \right]$$

and

$$\mathcal{M}_k = \mathbb{E}_{\mathcal{T}_m} \left[ \mathcal{L}_m(f_{\boldsymbol{\theta}}) - \sum_{i=0}^{k-1} \beta_i \right],$$

where $\beta_i = \alpha \nabla_{\boldsymbol{\theta}_i} \mathcal{L}_m(f_{\boldsymbol{\theta}_i}) \nabla_{\boldsymbol{\theta}} \mathcal{L}_m(f_{\boldsymbol{\theta}})^{\mathsf{T}}$ and $\boldsymbol{\theta}_0 = \boldsymbol{\theta}, \boldsymbol{\theta}_{i+1} = \boldsymbol{\theta}_i - \alpha \nabla_{\boldsymbol{\theta}_i} \mathcal{L}(f_{\boldsymbol{\theta}_i}, \mathcal{D}_m^{tr})$. The result is straightforward now.

■

## D  PROOF OF THEOREM 3

The proof techniques we use are similar to some previous works such as (Arora et al., 2019; Allen-Zhu et al., 2019). We summaries some of the differences. Different from previous works that typically assume a neural network is Gaussian initialized, we do not have such an assumption as we are trying to learn a good meta-initialization in the meta-learning setting. Previous works try to investigate the behavior of models during training, while we focus on revealing the connection between different meta-learning algorithms. Previous work focuses on single-task regression/classification problems, while we focus on meta-learning problem.

**Theorem 3** *Let $f_{\boldsymbol{\theta}}$ be a fully-connected neural network with $L$ hidden layers and ReLU activation function, $s_1, ..., s_{L+1}$ be the spectral norm of the weight matrices, $s = \max_h s_h$, and $\alpha$ be the learning rate of gradient descent. If $\alpha \le O(qr)$ with $q = \min(1/(Ls^L), L^{-1/(L+1)})$ and $r = \min(s^{-L}, s)$, then the following holds*

$$|\widetilde{\mathcal{E}}(k\alpha, f_{\boldsymbol{\theta}}) - \mathcal{M}_k| \le O\left(\frac{1}{L}\right).$$

**Proof** We first prove the case of $k = 2$, i.e. applying a two-step gradient descent adaptation in MAML.

We need to prove the following theorem first.

**Theorem 6** *Let $f_{\boldsymbol{\theta}}$ be a fully-connected neural network with $L$ hidden layers, and $\mathbf{x}$ be a data sample. Represent the neural network by $f_{\boldsymbol{\theta}}(\mathbf{x}) = \sigma(\sigma(...\sigma(\mathbf{x} W^1)...W^{L-1})W^L)W^{L+1}$, where $W^1, ..., W^{L+1}$ denote the weight matrices, and $\sigma$ is the ReLU activation function. Let $s_1, ..., s_{L+1}$ be the spectral norm of weight matrices, and $s = \max_h s_h$. Let $\alpha$ be the learning rate of gradient descent, and $f_{\tilde{\boldsymbol{\theta}}}(\mathbf{x})$ be the resulting value after one step of gradient descent, and $\| \cdot \|_{\mathcal{F}}$ be the Frobenius norm.*

*If $\alpha \le O(qs^{-L})$, where $q = \min(1/(Ls^L), L^{-1/(L+1)})$, then*

$$\left\| \frac{\partial f_{\tilde{\boldsymbol{\theta}}}(\mathbf{x})}{\partial \tilde{\boldsymbol{\theta}}} - \frac{\partial f_{\boldsymbol{\theta}}(\mathbf{x})}{\partial \boldsymbol{\theta}} \right\|_{\mathcal{F}} \le O\left(\frac{1}{s\sqrt{L+1}}\right).$$

**Remark 1** *Theorem 6 states that for a neural network with $L$ hidden layers, if the learning rate of gradient descent is bounded, then the norm of derivative w.r.t all the parameters will not change*

*too much, although there are $O(Lm^2)$ parameters, where $m$ denotes the maximum width of hidden layers. We use row vector instead of column vector for consistency, while it does not affect our results.*

For simplicity, we will write $g^h(\mathbf{x})$ as $g^h$. The bias terms in the neural network are introduced by adding an additional coordinate thus omitted in Theorem 6. Without loss of generality, we can assume $\|\mathbf{x}\| \leq 1$, which can be done by data normalization in pre-processing.

Let $g^h(\mathbf{x}) = \sigma(\sigma(...\sigma(\mathbf{x}\,W^1)...W^{h-1})W^h)$ be the activation at $h^{th}$ hidden layer and $g^0(\mathbf{x}) = \mathbf{x}, g^{L+1} = f_{\boldsymbol{\theta}}(\mathbf{x})$. Define diagonal matrices $D^h$, where $D^h_{(i,i)} = \mathbf{1}\{g^{h-1}W^h \geq 0\}$ and

$$b^h = \begin{cases} \mathbf{I}_{d_y}, & \text{if } h = L+1 \\ b^{h+1}(W^{h+1})^{\mathsf{T}}D^h, & \text{otherwise} \end{cases}$$

where $\mathbf{I}_{d_y}$ is a $d_y \times d_y$ identity matrix. We first prove the following Lemma.

**Lemma 7** *Given a neural network as stated in Theorem 6, let $\|\cdot\|_2$ denote the spectral norm, $\triangle W^h = \tilde{W}^h - W^h$ denote some perturbation on weight matrices, $\tilde{g}^h(\mathbf{x})$ denote the resulting value after perturbation, and $\triangle g^h(\mathbf{x}) = \tilde{g}^h(\mathbf{x}) - g^h(\mathbf{x})$. If $s \geq 1$ and $\|\triangle W^h\|_2 \leq O(s^{-L}/L)$ for all h, then*

$$\|\triangle g^h\| \leq O(\frac{1}{Ls^{L-h+1}});$$

*If $s < 1$ and $\|\triangle W^h\|_2 \leq O(q)$ for all h, where $q = \min(1/(Ls^L), L^{-1/(L+1)})$ and $r = \max(q, s)$, then*

$$\|\triangle g^h\| \leq O(r^{h-1}q) = \begin{cases} O(\frac{1}{Ls^{L-h+1}}), & \text{if } 1/(Ls^L) \leq L^{-1/(L+1)} \\ O(L^{-h/(L+1)}), & \text{if } 1/(Ls^L) > L^{-1/(L+1)}. \end{cases}$$

**Proof** Proof of Lemma 7 is based on induction.

We first prove the case of $s \geq 1$. Note that $g^0 = \mathbf{x}$, thus $\triangle g^0 = 0 \leq O(\frac{1}{Ls^{L-0+1}})$ always holds.

For $\triangle g^1$, we have

$$\begin{aligned} \|\triangle g^1\| &= \|\sigma(\mathbf{x}\,\tilde{W}^1) - \sigma(\mathbf{x}\,W^1)\| \\ &\leq \|\mathbf{x}\,\tilde{W}^1 - \mathbf{x}\,W^1\|, \quad \text{due to the property of ReLU activation} \\ &\leq \|\mathbf{x}\|\|\triangle W^1\|_2 \\ &\leq O(\frac{1}{Ls^L}). \end{aligned}$$

Thus, the hypothesis holds for $\triangle g^1$.

Now, assume that the hypothesis holds for $\triangle g^h$, then we have

$$\begin{aligned} \|\triangle g^{h+1}\| &= \|\sigma(\tilde{g}^h\tilde{W}^{h+1}) - \sigma(g^hW^{h+1})\| \\ &\leq \|\tilde{g}^h\tilde{W}^{h+1} - g^hW^{h+1}\|, \quad \text{due to the property of ReLU activation} \\ &\leq \|\tilde{g}^hW^{h+1} + \tilde{g}^h\triangle W^{h+1} - g^hW^{h+1}\| \\ &\leq \|\triangle g^h\|\|W^{h+1}\|_2 + \|\tilde{g}^h\|\|\triangle W^{h+1}\|_2 \\ &\leq O(s)\|\triangle g^h\| + \|g^h + \triangle g^h\|\|\triangle W^{h+1}\|_2 \\ &\leq O(s)\|\triangle g^h\| + O(s^h)\|\triangle W^{h+1}\|_2 + \|\triangle g^h\|\|\triangle W^{h+1}\|_2 \\ &\leq O(s)O(\frac{1}{Ls^{L-h+1}}) + O(s^h)O(\frac{1}{Ls^L}) + O(\frac{1}{Ls^{L-h+1}})O(\frac{1}{Ls^L}) \\ &\leq O(\frac{1}{Ls^{L-h}}). \end{aligned}$$

The last three inequalities come from the fact that $g^h = \sigma(\sigma(...\sigma(\mathbf{x}\,W^1)...W^{h-1})W^h) \leq O(s^h)$ and $s \geq 1$. Thus, we have proved the Lemma in the case $s \geq 1$.

Now, we prove the first part of the case of $s < 1$, i.e. $\|\triangle g^h\| \leq O(r^{h-1}q)$. Because $\triangle g^0 = 0$, thus the hypothesis for $\triangle g^0$ always holds.

For $\triangle g^1$, we have

$$
\begin{aligned}
\|\triangle g^1\| &= \|\sigma(\mathbf{x}\,\tilde{W}^1) - \sigma(\mathbf{x}\,W^1)\| \\
&\leq \|\mathbf{x}\,\tilde{W}^1 - \mathbf{x}\,W^1\| \\
&\leq \|\mathbf{x}\|\|\triangle W^1\|_2 \\
&\leq O(q).
\end{aligned}
$$

Thus, the hypothesis holds for $\triangle g^1$.

Now, we assume that the hypothesis holds for $\triangle g^h$. Then, we have

$$
\begin{aligned}
\|\triangle g^{h+1}\| &= \|\sigma(\tilde{g}^h \tilde{W}^{h+1}) - \sigma(g^h W^{h+1})\| \\
&\leq \|\tilde{g}^h \tilde{W}^{h+1} - g^h W^{h+1}\| \\
&\leq \|\tilde{g}^h W^{h+1} + \tilde{g}^h \triangle W^{h+1} - g^h W^{h+1}\| \\
&\leq \|\triangle g^h\|\|W^{h+1}\|_2 + \|\tilde{g}^h\|\|\triangle W^{h+1}\|_2 \\
&\leq O(s)\|\triangle g^h\| + \|g^h + \triangle g^h\|\|\triangle W^{h+1}\|_2 \\
&\leq O(s)O(r^{h-1}q) + O(s^h)q + qO(r^{h-1}q) \\
&\leq O(r^h q).
\end{aligned}
$$

The last inequality comes from the fact that $r = \max(q, s)$ and $s^h < s < 1$.

Next we consider the second part of the case of $s < 1$.

If $1/(Ls^L) \leq L^{-1/(L+1)}$, we know that $q = 1/(Ls^L)$ and

$$
\begin{aligned}
1/(Ls^L) &\leq L^{-1/(L+1)} \\
L^{1/(L+1)} &\leq Ls^L \\
L^{-L/(L+1)} &\leq s^L \\
L^{-1} &\leq s^{L+1} \\
L^{-1}s^{-L} &\leq s,
\end{aligned}
$$

which means $q \leq s$, thus $r = s$. Then, we have

$$
\|\triangle g^h\| = O(r^{h-1}q) = O(s^{h-1}q) = O(s^{h-1}L^{-1}s^{-L}) = O(\frac{1}{Ls^{L-h+1}}).
$$

If $1/(Ls^L) > L^{-1/(L+1)}$, we know that $q = L^{-1/(L+1)}$ and $q > s$; then, $r = q$ and

$$
\|\triangle g^h\| = O(r^{h-1}q) = O(q^{h-1}q) = O(q^h) = O(L^{-h/(L+1)}).
$$

Thus, we can conclude that Lemma 7 also holds for the case of $s < 1$, which completes the proof. ∎

We now prove a similar Lemma for $\triangle b^h$.

**Lemma 8** *Given a neural network as stated in Theorem 6, let $\|\cdot\|_2$ denote the spectral norm, $\triangle W^h = \tilde{W}^h - W^h$ denote some perturbation on weight matrices, $\tilde{b}^h$ denote the resulting value after perturbation, and $\triangle b^h = \tilde{b}^h - b^h$.*

*If $s \geq 1$ and $\|\triangle W^h\|_2 \leq O(s^{-L}/L)$ for all h, then*

$$
\|\triangle b^h\| \leq O(\frac{1}{Ls^h});
$$

*If $s < 1$ and $\|\triangle W^h\|_2 \leq O(q)$ for all h, where $q = \min(1/(Ls^L), L^{-1/(L+1)})$, then*

$$
\|\triangle b^h\| \leq \begin{cases} O(L^{-1}s^{-h}), & \text{if } 1/(Ls^L) \leq L^{-1/(L+1)} \\ O(L^{(h-L-1)/(L+1)}), & \text{if } 1/(Ls^L) > L^{-1/(L+1)}. \end{cases}
$$

**Proof** Recall that

$$b^h = \begin{cases} \mathbf{I}_{d_y}, & \text{if } h = L+1 \\ b^{h+1}(W^{h+1})^\intercal D^h, & \text{otherwise} \end{cases}$$

where $\mathbf{I}_{d_y}$ is a $d_y \times d_y$ identity matrix and $D^h_{(i,i)} = \mathbf{1}\{g^{h-1}W^h \geq 0\}$. It is easy to see that $\|b^h\| \leq O(s^{L-h+1})$, because $\|D^h\|_2 \leq 1$ and $\|W^h\|_2 \leq s$.

We first prove the case of $s \geq 1$. We know that $\triangle b^{L+1} = 0 \leq O(s^{-L-1}/L)$ always holds.

For $h \leq L$, we can re-write $b^h$ as

$$b^h = \mathbf{I}_{d_y}(W^{L+1})^\intercal D^L(W^L)^\intercal D^{L-1}...(W^{h+1})^\intercal D^h.$$

Then, we have

$$b^h(g^h)^\intercal = \mathbf{I}_{d_y}(W^{L+1})^\intercal D^L(W^L)^\intercal D^{L-1}...(W^{h+1})^\intercal D^h(g^h)^\intercal. \tag{10}$$

Because of the fact that

$$f_{\boldsymbol{\theta}} = g^{L+1} = \mathbf{x}\, W^1 D^1 W^2 D^2 ... D^L W^{L+1} = g^h W^{h+1} D^{h+1} ... D^L W^{L+1}$$

and $g^h = g^h D^h$, $D^h = (D^h)^\intercal$. We can re-write equation 10 as

$$b^h(g^h)^\intercal = f_{\boldsymbol{\theta}}^\intercal.$$

Thus,

$$\|\tilde{b}^h(\tilde{g}^h)^\intercal - b^h(g^h)^\intercal\| = \|\boldsymbol{f}_{\tilde{\boldsymbol{\theta}}} - f_{\boldsymbol{\theta}}\| = \|\triangle g^{L+1}\| \leq O(\frac{1}{L})$$

by Lemma 7. Consequently, we have

$$\|\tilde{b}^h(\tilde{g}^h)^\intercal - b^h(g^h)^\intercal\| = \|\triangle b^h(g^h)^\intercal + \triangle b^h \triangle(g^h)^\intercal + \tilde{b}^h \triangle(g^h)^\intercal\| \leq O(\frac{1}{L}).$$

Since $\|g^h\| \leq O(s^h)$, we know that

$$\|\triangle b^h\| \leq O(\frac{1}{Ls^h}), \quad \|\triangle b^h\| \leq O(s^{L-h+1})$$

always hold. Since $L \geq 1, s \geq 1$, we simply have $\|\triangle b^h\| \leq O(\frac{1}{Ls^h})$.

Now, we prove the case of $s < 1$. Similarly, we have

$$\|\tilde{b}^h(\tilde{g}^h)^\intercal - b^h(g^h)^\intercal\| = \|\boldsymbol{f}_{\tilde{\boldsymbol{\theta}}} - f_{\boldsymbol{\theta}}\| = \|\triangle g^{L+1}\| \leq O(\frac{1}{L}).$$

Similarly, we must have

$$\|\triangle b^h\| \leq O(\frac{1}{Ls^h}), \quad \|\triangle b^h\| \leq O(\frac{1}{Lr^{h-1}q}),$$

where $q = \min(1/(Ls^L), L^{-1/(L+1)})$ and $r = \max(q, s)$ by Lemma 7.

If $1/(Ls^L) \leq L^{-1/(L+1)}$, then $s^{L+1} \geq 1/L$. We thus have

$$O(\frac{1}{Lr^{h-1}q}) = O(\frac{Ls^{L-h+1}}{L}) = O(\frac{s^{L+1}}{s^h}) \geq O(\frac{1}{Ls^h}).$$

Hence, we get $\|\triangle b^h\| \leq O(\frac{1}{Ls^h})$.

If $1/(Ls^L) > L^{-1/(L+1)}$, then $s^{L+1} < 1/L$. We have

$$O(\frac{1}{Lr^{h-1}q}) = O(L^{-1} \cdot L^{h/(L+1)}) \leq O(L^{-1} \cdot s^{-h}) = O(\frac{1}{Ls^h}).$$

Thus, we get $\|\triangle b^h\| \leq O(L^{(h-L-1)/(L+1)})$. ∎

**Lemma 9** *Given a neural network as stated in Theorem 6, let $\| \cdot \|_\mathcal{F}$ be the Frobenius norm, $W^1, ..., W^{L+1}$ be the weight matrices in the neural network, $\triangle W^h = \tilde{W}^h - W^h$ be the perturbation on weight matrices, $\boldsymbol{\theta}^h$ be the parameter vector containing all the elements in $W^h$, $\triangle \boldsymbol{\theta}^h = \tilde{\boldsymbol{\theta}}^h - \boldsymbol{\theta}^h$ be the perturbation on parameter vectors, and $\boldsymbol{f}_{\tilde{\boldsymbol{\theta}}}(\mathbf{x})$ be the resulting value after perturbation.*

*If $s \geq 1$ and $\|\triangle W^h\|_2 \leq O(s^{-L}/L)$ for all h, for any weight matrices the following holds*

$$\Big\| \frac{\partial \boldsymbol{f}_{\tilde{\boldsymbol{\theta}}}(\mathbf{x})}{\partial \tilde{\boldsymbol{\theta}}^h} - \frac{\partial f_{\boldsymbol{\theta}}(\mathbf{x})}{\partial \boldsymbol{\theta}^h} \Big\|_\mathcal{F} \leq O(\frac{1}{sL});$$

*If $s < 1$ and $\|\triangle W^h\|_2 \leq O(q)$ for all h, where $q = \min(1/(Ls^L), L^{-1/(L+1)})$, for any weight matrices the following holds*

$$\Big\| \frac{\partial \boldsymbol{f}_{\tilde{\boldsymbol{\theta}}}(\mathbf{x})}{\partial \tilde{\boldsymbol{\theta}}^h} - \frac{\partial f_{\boldsymbol{\theta}}(\mathbf{x})}{\partial \boldsymbol{\theta}^h} \Big\|_\mathcal{F} \leq O(\frac{1}{sL}).$$

**Proof** We first prove the case of $d_y = 1$, i.e. the output of neural network is 1-dimensional.

In this case, we know that

$$\Big\| \frac{\partial \boldsymbol{f}_{\tilde{\boldsymbol{\theta}}}(\mathbf{x})}{\partial \tilde{\boldsymbol{\theta}}^h} - \frac{\partial f_{\boldsymbol{\theta}}(\mathbf{x})}{\partial \boldsymbol{\theta}^h} \Big\|_\mathcal{F} = \Big\| \frac{\partial \boldsymbol{f}_{\tilde{\boldsymbol{\theta}}}(\mathbf{x})}{\partial \tilde{W}^h} - \frac{\partial f_{\boldsymbol{\theta}}(\mathbf{x})}{\partial W^h} \Big\|_\mathcal{F} = \Big\| \triangle \frac{\partial f_{\boldsymbol{\theta}}(\mathbf{x})}{\partial W^h} \Big\|_\mathcal{F}$$

and the derivative to $W^h$ is

$$\frac{\partial f_{\boldsymbol{\theta}}(\mathbf{x})}{\partial W^h} = (b^h)^\intercal g^{h-1}.$$

Then, we have

$$\begin{aligned}
\Big\| \triangle \frac{\partial f_{\boldsymbol{\theta}}(\mathbf{x})}{\partial W^h} \Big\|_\mathcal{F} &= \|(\tilde{b}^h)^\intercal \tilde{g}^{h-1} - (b^h)^\intercal g^{h-1}\|_\mathcal{F} \\
&= \|(\tilde{b}^h)^\intercal g^{h-1} - (b^h)^\intercal g^{h-1} + (\tilde{b}^h)^\intercal \triangle g^{h-1}\|_\mathcal{F} \\
&\leq \|(\triangle b^h)^\intercal g^{h-1}\|_\mathcal{F} + \|(b^h + \triangle b^h)^\intercal \triangle g^{h-1}\|_\mathcal{F}.
\end{aligned}$$

Recall the fact that $g^h \leq O(s^h)$ and $b^h \leq O(s^{L+1-h})$.

When $s \geq 1$, from Lemma 7 and Lemma 8 we know that

$$\|\triangle g^h\| \leq O(\frac{1}{Ls^{L-h+1}}), \quad \|\triangle b^h\| \leq O(\frac{1}{Ls^h}).$$

Then, we have

$$\begin{aligned}
\Big\| \triangle \frac{\partial f_{\boldsymbol{\theta}}(\mathbf{x})}{\partial W^h} \Big\|_\mathcal{F} &\leq O(s^{h-1})O(\frac{1}{Ls^h}) + O(s^{L+1-h})O(\frac{1}{Ls^{L-h+2}}) + O(\frac{1}{Ls^{L-h+2}})O(\frac{1}{Ls^h}) \\
&\leq O(\frac{1}{sL}).
\end{aligned}$$

When $s < 1$, from Lemma 7 and Lemma 8 we know that

$$\|\triangle g^h\| \leq \begin{cases} O(\frac{1}{Ls^{L-h+1}}), & \text{if } 1/(Ls^L) \leq L^{-1/(L+1)} \\ O(L^{-h/(L+1)}), & \text{if } 1/(Ls^L) > L^{-1/(L+1)} \end{cases}$$

and

$$\|\triangle b^h\| \leq \begin{cases} O(L^{-1}s^{-h}), & \text{if } 1/(Ls^L) \leq L^{-1/(L+1)} \\ O(L^{(h-L-1)/(L+1)}), & \text{if } 1/(Ls^L) > L^{-1/(L+1)}. \end{cases}$$

If $1/(Ls^L) \leq L^{-1/(L+1)}$, we have

$$\Big\| \triangle \frac{\partial f_{\boldsymbol{\theta}}(\mathbf{x})}{\partial W^h} \Big\|_\mathcal{F} \leq O(s^{h-1})O(\frac{1}{Ls^h}) + O(s^{L-h+1})O(\frac{1}{Ls^{L-h+2}}) + O(\frac{1}{Ls^{L-h+2}})O(\frac{1}{Ls^h}).$$

Since $1/(Ls^L) \leq L^{-1/(L+1)}$ implies $L^{-1} \leq s^{L+1}$ (from proof of Lemma 7), we have

$$\frac{1}{Ls^h} \leq s^{L-h+1}.$$

Then we can conclude that

$$\left\|\triangle\frac{\partial f_{\boldsymbol{\theta}}(\mathbf{x})}{\partial W^h}\right\|_{\mathcal{F}} \leq O(\frac{1}{sL}).$$

If $1/(Ls^L) > L^{-1/(L+1)}$, we have

$$\left\|\triangle\frac{\partial f_{\boldsymbol{\theta}}(\mathbf{x})}{\partial W^h}\right\|_{\mathcal{F}} \leq O(s^{h-1})O(L^{(h-L-1)/(L+1)}) + O(s^{L+1-h})O(L^{-(h-1)/(L+1)})$$
$$+ O(L^{-(h-1)/(L+1)})O(L^{(h-L-1)/(L+1)}).$$

Since $1/(Ls^L) > L^{-1/(L+1)}$ implies $L^{-1} > s^{L+1}$ (from proof of Lemma 7), we have

$$L^{(h-L-1)/(L+1)} > s^{L-h+1}, \quad \frac{1}{L^{(h-1)/(L+1)}} > s^{h-1}.$$

Then we have

$$\left\|\triangle\frac{\partial f_{\boldsymbol{\theta}}(\mathbf{x})}{\partial W^h}\right\|_{\mathcal{F}} \leq O(\frac{1}{L}) \leq O(\frac{1}{sL}), \text{ because } s < 1.$$

We have proved the Lemma for the case of $d_y = 1$.

For the case of $d_y > 1$, we know that

$$\left\|\frac{\partial \boldsymbol{f}_{\tilde{\boldsymbol{\theta}}}(\mathbf{x})}{\partial \tilde{\boldsymbol{\theta}}^h} - \frac{\partial f_{\boldsymbol{\theta}}(\mathbf{x})}{\partial \boldsymbol{\theta}^h}\right\|_{\mathcal{F}}^2 = \sum_{i=1}^{d_y}\left\|\frac{\partial \boldsymbol{f}_{\tilde{\boldsymbol{\theta}},i}(\mathbf{x})}{\partial \tilde{\boldsymbol{\theta}}^h} - \frac{\partial f_{\boldsymbol{\theta},i}(\mathbf{x})}{\partial \boldsymbol{\theta}^h}\right\|_{\mathcal{F}}^2 \leq O(\frac{d_y}{s^2L^2}),$$

where $f_{\boldsymbol{\theta},i}(\mathbf{x})$ is the $i^{th}$ dimension of $f_{\boldsymbol{\theta}}(\mathbf{x})$. The last inequality directly comes from the 1-dimensional case.

Since $d_y$ is a constant, we ignore it. Then, we have

$$\left\|\frac{\partial \boldsymbol{f}_{\tilde{\boldsymbol{\theta}}}(\mathbf{x})}{\partial \tilde{\boldsymbol{\theta}}^h} - \frac{\partial f_{\boldsymbol{\theta}}(\mathbf{x})}{\partial \boldsymbol{\theta}^h}\right\|_{\mathcal{F}} \leq O(\frac{1}{sL}),$$

which completes the proof. ∎

Now we can prove Theorem 6, if $\tilde{W}^h$ is obtained by one step gradient descent starting from $W^h$, $\tilde{\boldsymbol{\theta}}$ is obtained by one step gradient descent starting from $\boldsymbol{\theta}$, and learning rate is $\alpha$. Then, for any weight matrix we have

$$\begin{aligned}
\|\triangle W^h\|_2 &= \|\alpha\nabla_{W^h}\mathcal{L}(\boldsymbol{\theta})\|_2 \\
&\leq \|\alpha\nabla_{W^h}\mathcal{L}(\boldsymbol{\theta})\|_{\mathcal{F}} \\
&= \|\alpha\nabla_{\boldsymbol{\theta}^h}\mathcal{L}(\boldsymbol{\theta})\|_{\mathcal{F}} \\
&= \alpha\left\|\frac{\sum_{i=1}^n[f_{\boldsymbol{\theta}}(\mathbf{x}_i) - \mathbf{y}_i]}{n}\frac{\partial f_{\boldsymbol{\theta}}(\mathbf{x}_i)}{\partial \boldsymbol{\theta}^h}^{\intercal}\right\|_{\mathcal{F}} \\
&\leq \frac{\alpha\sum_{i=1}^n}{n}c_i\left[\sum_j^{d_y}\left\|\frac{\partial f_{\boldsymbol{\theta},j}(\mathbf{x}_i)}{\partial W^h}\right\|_{\mathcal{F}}^2\right]^{1/2} \\
&\leq \frac{\alpha\sum_{i=1}^n}{n}c_i\sqrt{d_y}O(s^{L-h+1})O(s^{h-1}) \\
&\leq \alpha O(s^L),
\end{aligned}$$

where $c_i = \|f_{\boldsymbol{\theta}}(\mathbf{x}_i) - \mathbf{y}_i\|$ are some constants.

If $\alpha \leq O(s^{-2L}/L)$ when $s \geq 1$, then for any weight matrix we have

$$\|\triangle W^h\|_2 \leq \alpha O(s^L) \leq O(s^{-L}/L).$$

If $\alpha \leq O(qs^{-L})$ where $q = \min(1/(Ls^L), L^{-1/(L+1)})$ when $s < 1$, then for any weight matrix we have

$$\|\triangle W^h\|_2 \leq \alpha O(s^L) \leq O(q).$$

By Lemma 9, we can conclude that

$$\left\|\frac{\partial \boldsymbol{f}_{\tilde{\boldsymbol{\theta}}}(\mathbf{x})}{\partial \tilde{W}^h} - \frac{\partial f_{\boldsymbol{\theta}}(\mathbf{x})}{\partial W^h}\right\|_{\mathcal{F}} \leq O(\frac{1}{sL}).$$

Then, we have

$$\left\|\frac{\partial \boldsymbol{f}_{\tilde{\boldsymbol{\theta}}}(\mathbf{x})}{\partial \tilde{\boldsymbol{\theta}}} - \frac{\partial f_{\boldsymbol{\theta}}(\mathbf{x})}{\partial \boldsymbol{\theta}}\right\|_{\mathcal{F}} = \left[\sum_{h=1}^{L+1}\left\|\frac{\partial \boldsymbol{f}_{\tilde{\boldsymbol{\theta}}}(\mathbf{x})}{\partial \tilde{\boldsymbol{\theta}}^h} - \frac{\partial f_{\boldsymbol{\theta}}(\mathbf{x})}{\partial \boldsymbol{\theta}^h}\right\|_{\mathcal{F}}^2\right]^{1/2}$$

$$\leq O(\frac{1}{s\sqrt{L+1}}).$$

When $s \geq 1$, we know that

$$s^{-L} \leq 1 \leq L^{L/(L+1)}.$$

Then, we have

$$\frac{1}{Ls^L} \leq \frac{1}{L} \leq L^{-1/(L+1)}.$$

Thus, we know $1/(Ls^L) = \min(1/(Ls^L), L^{-1/(L+1)})$ when $s \geq 1$.

For the case of $s \geq 1$, we can rewrite $\alpha \leq O(s^{-2L}/L) = O(qs^{-L})$, where $q = \min(1/(Ls^L), L^{-1/(L+1)})$, which completes the proof of Theorem 6.

Now, we prove Theorem 3 with $k = 2$, i.e. two-step gradient descent adaptation. We know that

$$\beta_1 = \alpha \nabla_{\tilde{\boldsymbol{\theta}}} \mathcal{L}_m(f_{\tilde{\boldsymbol{\theta}}}) \nabla_{\boldsymbol{\theta}} \mathcal{L}_m(f_{\boldsymbol{\theta}})^{\mathsf{T}}, \|\nabla_{f_{\boldsymbol{\theta}}} \mathcal{L}_m(f_{\boldsymbol{\theta}})\|_{\mathcal{H}}^2 = \|\nabla_{\boldsymbol{\theta}} \mathcal{L}_m(f_{\boldsymbol{\theta}})\|^2.$$

Thus, we have

$$\left|\beta_1 - \alpha\|\nabla_{f_{\boldsymbol{\theta}}} \mathcal{L}_m(f_{\boldsymbol{\theta}})\|_{\mathcal{H}}^2\right|$$

$$=\left|\alpha \nabla_{\tilde{\boldsymbol{\theta}}} \mathcal{L}_m(f_{\tilde{\boldsymbol{\theta}}}) \nabla_{\boldsymbol{\theta}} \mathcal{L}_m(f_{\boldsymbol{\theta}})^{\mathsf{T}} - \alpha \nabla_{\boldsymbol{\theta}} \mathcal{L}_m(f_{\boldsymbol{\theta}}) \nabla_{\boldsymbol{\theta}} \mathcal{L}_m(f_{\boldsymbol{\theta}})^{\mathsf{T}}\right|$$

$$=\alpha\|\nabla_{\tilde{\boldsymbol{\theta}}} \mathcal{L}_m(f_{\tilde{\boldsymbol{\theta}}}) - \nabla_{\boldsymbol{\theta}} \mathcal{L}_m(f_{\boldsymbol{\theta}})\|\|\nabla_{\boldsymbol{\theta}} \mathcal{L}_m(f_{\boldsymbol{\theta}})\|$$

$$=\alpha\left\|\mathbb{E}_{(\mathbf{x}_m, \mathbf{y}_m)}\left\{[f_{\tilde{\boldsymbol{\theta}}}(\mathbf{x}_m) - \mathbf{y}_m]\frac{\partial f_{\tilde{\boldsymbol{\theta}}}(\mathbf{x}_m)}{\partial \tilde{\boldsymbol{\theta}}}^{\mathsf{T}} - [f_{\boldsymbol{\theta}}(\mathbf{x}_m) - \mathbf{y}_m]\frac{\partial f_{\boldsymbol{\theta}}(\mathbf{x}_m)}{\partial \boldsymbol{\theta}}^{\mathsf{T}}\right\}\right\|\|\nabla_{\boldsymbol{\theta}} \mathcal{L}_m(f_{\boldsymbol{\theta}})\|$$

$$=\alpha\left\|\mathbb{E}_{(\mathbf{x}_m, \mathbf{y}_m)}\left\{[f_{\boldsymbol{\theta}}(\mathbf{x}_m) - \mathbf{y}_m + \triangle f_{\boldsymbol{\theta}}(\mathbf{x}_m)]\left[\frac{\partial f_{\tilde{\boldsymbol{\theta}}}(\mathbf{x}_m)}{\partial \tilde{\boldsymbol{\theta}}} + \triangle\frac{\partial f_{\boldsymbol{\theta}}(\mathbf{x}_m)}{\partial \boldsymbol{\theta}}\right]^{\mathsf{T}}\right.\right.$$

$$\left.\left. - [f_{\boldsymbol{\theta}}(\mathbf{x}_m) - \mathbf{y}_m]\frac{\partial f_{\boldsymbol{\theta}}(\mathbf{x}_m)}{\partial \boldsymbol{\theta}}^{\mathsf{T}}\right\}\right\|\|\nabla_{\boldsymbol{\theta}} \mathcal{L}_m(f_{\boldsymbol{\theta}})\|$$

$$=\alpha\left\|\mathbb{E}_{(\mathbf{x}_m, \mathbf{y}_m)}\left\{\triangle f_{\boldsymbol{\theta}}(\mathbf{x}_m)\left[\frac{\partial f_{\tilde{\boldsymbol{\theta}}}(\mathbf{x}_m)}{\partial \tilde{\boldsymbol{\theta}}} + \triangle\frac{\partial f_{\boldsymbol{\theta}}(\mathbf{x}_m)}{\partial \boldsymbol{\theta}}\right]^{\mathsf{T}}\right.\right.$$

$$\left.\left. + [f_{\boldsymbol{\theta}}(\mathbf{x}_m) - \mathbf{y}_m]\triangle\frac{\partial f_{\boldsymbol{\theta}}(\mathbf{x}_m)}{\partial \boldsymbol{\theta}}^{\mathsf{T}}\right\}\right\|\|\nabla_{\boldsymbol{\theta}} \mathcal{L}_m(f_{\boldsymbol{\theta}})\|$$

$$\leq\alpha\left[O(\frac{1}{L})O(s^L\sqrt{L}) + O(\frac{1}{L})O(\frac{1}{s\sqrt{L}}) + O(\frac{1}{s\sqrt{L}})\right]\|\nabla_{\boldsymbol{\theta}} \mathcal{L}_m(f_{\boldsymbol{\theta}})\|$$

$$\leq\alpha\left[O(\frac{s^L}{\sqrt{L}}) + O(\frac{1}{s\sqrt{L}})\right]\|\nabla_{\boldsymbol{\theta}} \mathcal{L}_m(f_{\boldsymbol{\theta}})\|, \text{ because } L \geq 1$$

$$\leq\left[O(\frac{qrs^L}{\sqrt{L}}) + O(\frac{qr}{s\sqrt{L}})\right]\|\nabla_{\boldsymbol{\theta}} \mathcal{L}_m(f_{\boldsymbol{\theta}})\|, \text{ where } q = \min(1/(Ls^L), L^{-1/(L+1)}), r = \min(s^{-L}, s)$$

$$\leq O(\frac{q}{\sqrt{L}})\|\nabla_{\boldsymbol{\theta}} \mathcal{L}_m(f_{\boldsymbol{\theta}})\|.$$

In the case of $d_y = 1$, we have

$$\left\|\frac{\partial f_{\boldsymbol{\theta}}(\mathbf{x})}{\partial W^h}\right\|_{\mathcal{F}} = (b^h)^{\mathsf{T}} g^{h-1} \leq O(s^L),$$

which has already been shown in the proof of Lemma 9. Then, we have

$$\|\nabla_{\boldsymbol{\theta}} \mathcal{L}_m(f_{\boldsymbol{\theta}})\| = O\left(\sqrt{\sum_{h=1}^{L+1} \left\|\frac{\partial f_{\boldsymbol{\theta}}(\mathbf{x})}{\partial \boldsymbol{\theta}^h}\right\|^2}\right) = O\left(\sqrt{\sum_{h=1}^{L+1} \left\|\frac{\partial f_{\boldsymbol{\theta}}(\mathbf{x})}{\partial W^h}\right\|_{\mathcal{F}}^2}\right) \leq O(s^L \sqrt{L+1}).$$

In the case of $d_y \geq 1$, the bound is simply scaled by a constant of $\sqrt{d_y}$.

Thus we have

$$\left|\beta_1 - \alpha\|\nabla_{f_{\boldsymbol{\theta}}} \mathcal{L}_m(f_{\boldsymbol{\theta}})\|_{\mathcal{H}}^2\right| \leq O\left(\frac{q}{\sqrt{L}}\right)\|\nabla_{\boldsymbol{\theta}} \mathcal{L}_m(f_{\boldsymbol{\theta}})\| \leq O(qs^L) \leq O\left(\frac{1}{L}\right)$$

because $q = \min(1/(Ls^L), L^{-1/(L+1)})$, which completes the proof for the case of $k = 2$.

For the case of $k > 2$, we only need to make sure that the bound on learning rate always holds. Fortunately, since $k$ is a finite constant, according to what we have already showed in the proof of previous lemmas, every step of gradient descent will not change the spectral norm of the weight matrix too much: $\|\triangle W^h\|_2 \leq O(s^{-L}/L)$ for all h if $s \geq 1$, and $\|\triangle W^h\|_2 \leq O(q)$ for all h if $s < 1$, where $q = \min(1/(Ls^L), L^{-1/(L+1)})$. Thus, we may assume that the bound on learning rate always holds during the adaptation. Using triangle inequality to generalize the results from $k = 2$ to $k > 2$, i.e. for all $1 \leq i \leq k - 1$, we have

$$\left|\beta_i - \alpha\|\nabla_{f_{\boldsymbol{\theta}}} \mathcal{L}_m(f_{\boldsymbol{\theta}})\|_{\mathcal{H}}^2\right| \leq O\left(\frac{1}{L}\right).$$

Recall that

$$\widetilde{\mathcal{E}}(\alpha, f_{\boldsymbol{\theta}}) = \mathbb{E}_{\mathcal{T}_m}\left[\mathcal{L}_m(f_{\boldsymbol{\theta}}) - \alpha\|\nabla_{f_{\boldsymbol{\theta}}} \mathcal{L}_m(f_{\boldsymbol{\theta}})\|_{\mathcal{H}}^2\right]$$

and

$$\mathcal{M}_k = \mathbb{E}_{\mathcal{T}_m}\left[\mathcal{L}_m(f_{\boldsymbol{\theta}}) - \sum_{i=0}^{k-1} \beta_i\right],$$

where $\beta_i = \alpha\nabla_{\boldsymbol{\theta}_i} \mathcal{L}_m(f_{\boldsymbol{\theta}_i})\nabla_{\boldsymbol{\theta}} \mathcal{L}_m(f_{\boldsymbol{\theta}})^{\mathsf{T}}$ and $\boldsymbol{\theta}_0 = \boldsymbol{\theta}, \boldsymbol{\theta}_{i+1} = \boldsymbol{\theta}_i - \alpha\nabla_{\boldsymbol{\theta}_i}\mathcal{L}(f_{\boldsymbol{\theta}_i}, \mathcal{D}_m^{tr})$. The result is straightforward now. ∎

## E    PROOF OF THEOREM 4

**Theorem 4** *Let $f_{\boldsymbol{\theta}}$ be a convolutional neural network with $L - l$ convolutional layers and $l$ fully-connected layers and with ReLU activation function, and $d_x$ be the input dimension. Denote by $W^h$ the parameter **vector** of the convolutional layer for $h \leq L - l$, and the weight **matrices** of the fully connected layers for $L - l + 1 < h \leq L + 1$. $\| \cdot \|_2$ means both the spectral norm of a matrix and the Euclidean norm of a vector. Define*

$$s_h = \begin{cases} \sqrt{d_x}\|W^h\|_2, & \text{if } h = 1, ..., L - l \\ \|W^h\|_2, & \text{if } L - l + 1 < h \leq L + 1 \end{cases}$$

*and let $s = \max_h s_h$ and $\alpha$ be the learning rate of gradient descent. If $\alpha \leq O(qr)$ with $q = \min(1/(Ls^L), L^{-1/(L+1)})$ and $r = \min(s^{-L}, s)$, the following holds*

$$|\mathcal{M}_k - \widetilde{\mathcal{E}}(k\alpha, f_{\boldsymbol{\theta}})| \leq O\left(\frac{1}{L}\right).$$

**Proof** We prove Theorem 4 by first transforming the convolutional neural network into an equivalent fully connected neural network and then applying Theorem 3.

First of all, we assume that there are $c_h$ channels in $h^{th}$ convolutional layer's output $g^h(\mathbf{x})$, where $h = 0, ..., L - l$. For fully-connected layers, define $c_{L-l} = ... = c_{L+1} = 1$. We may represent the dimensionality of input data by $\mathbf{x} \in R^{d_x c_0}$. Instead of using matrices, we represent the output of every convolutional layer by a $d_x c_h$ length vector $g^h = [g_1^h, g_2^h, ..., g_{d_x}^h]$, where every $g_i^h = [g_{i,1}^h, g_{i,2}^h, ..., g_{i,c_h}^h]$ is a $c_h$ length vector contains value of different channels at the same position.

We assume that for every element $g_{i,j}^h$ of $g_i^h$, its value is completely determined by elements of set $Q_i^{h-1}$, where $Q_i^{h-1}$ contains $kc_{h-1}$ elements with fixed positions in $g^{h-1}$ for a given $i$. In other words, every element of the output of a convolutional layer is determined by some elements with fixed positions from output of the previous layer. This is exactly how convolutional layer works in deep learning.

If we use $g_{Q_i^{h-1}}^{h-1}$ to represent the concatenation of $g_{a,b}^{h-1} \in Q_i^{h-1}$, then $g_{Q_i^{h-1}}^{h-1}$ is a $kc_{h-1}$ length vector, where $k$ is the kernel size. Then we have

$$g_i^h = \sigma(g_{Q_i^{h-1}}^{h-1} U_i^h)$$

where $U_{i,j}^h \in R^{kc_{h-1} \times c_h}$ is a $kc_{h-1} \times c_h$ matrix.

For notation simplicity, one can define a matrix $U^h \in R^{d_x c_{h-1} \times d_x c_h}$, where every column of $U^h$ only has $kc_{h-1}$ non-zero elements, and it satisfies

$$g^h = \sigma(g^{h-1} U^h)$$

By the property of convolutional layer, we know the following facts:
- One can represent $U^h$ by $U^h = [V_1^h, V_2^h, ..., V_{d_x}^h]$ where $V_i^h \in R^{d_x c_{h-1} \times c_h}$ is sub-matrix of $U^h$;
- Every $V_i^h$ contains the same set of elements as $W^h$, while these elements are located at different positions;
- Every $V_i^h$ can be obtained by any other $V_j^h$ by swapping rows;

Let's define $U^{L-l} = W^{L-l}, ..., U^{L+1} = W^{L+1}$ for the fully-connected layer and output layer. Then we can represent the neural network just as in Theorem 3 by $f_\theta(\mathbf{x}) = \sigma(\sigma(...\sigma(\mathbf{x} U^1)...U^{L-1})U^L)U^{L+1}$, and $\mathbf{x} \in R^{d_x c_0}$.

Now let $t_h$ be the spectral norm of $U^h$, and $t = \max_h t_h$. By Theorem 3, we know that we want $\alpha \leq O(qr)$, where $q = \min(1/(Ls^L), L^{-1/(L+1)}), r = \min(s^{-L}, s)$.

Because every $V_i^h$ contains the same set of elements, we know that every $V_i^h$ has the same Frobenius norm. Because every $V_i^h$ can be obtained by any other $V_j^h$ by swapping rows, we know that every $V_i^h$ has the same rank.

We know that

$$\frac{1}{\sqrt{r}} \|V_1^h\|_\mathcal{F} \leq \|V_1^h\|_2 \leq \|U^h\|_2 \leq \|U^h\|_\mathcal{F} = \sqrt{d_x} \|V_1^h\|_\mathcal{F} = \sqrt{d_x} \|W^h\|_2$$

where $\|\cdot\|_\mathcal{F}$ denotes Frobenius norm, $r$ denotes the rank of $V_1^h$. The last equality holds because matrix $V_1^h$ and vector $W^h$ have the same set of elements.

Let's define

$$s_h = \begin{cases} \sqrt{d_x} \|W^h\|_2, & \text{if } h = 1, ..., L - l \\ \|W^h\|_2, & \text{if } L - l + 1 < h \leq L + 1 \end{cases}$$

and $s = \max_h s_h$.

From above we know that $t_h = \Theta(s_h)$, because $s_h/\sqrt{d_x r} \leq t_h \leq s_h$. So we also have $t = \Theta(s)$. Then the conclusion is straightforward. ∎

## F    REVISION OF THEOREM 3 AND THEOREM 4 IN CLASSIFICATION CASE

We now show how to obtain similar results of Theorem 3 and Theorem 4 in classification problem, where cross-entropy loss is used instead of squared loss. We need two more restrictions in the classification case:

1. There exist matrix $A$ and $B$ such that $g^L A \leq \text{softmax}(g^L W^{L+1}) \leq g^L B$ for all data points, where softmax is the softmax operation at the last layer.

2. For any data point $\mathbf{x}$ whose belongs to $c^{th}$ class, there exists a constant $\epsilon > 0$ such that $f_{\boldsymbol{\theta},c}(\mathbf{x}) \geq \epsilon$, i.e. the output of neural network has a lower bound on the true class position.

The proof is actually similar to the proof in regression case. We briefly talk about the differences here.

Firstly, in the classification case, softmax function is used at the last layer. By the first restriction, we can get rid of softmax function by introducing new matrices, which further leads to bound of the learning rate as in regression case.

Secondly, if the loss function is the cross-entropy loss, we have:

$$\nabla_{\boldsymbol{\theta}} \mathcal{L}_m(f_{\boldsymbol{\theta}}) = \mathbb{E}_{(\mathbf{x}_m, \mathbf{y}_m)} \left[ \frac{1}{f_{\boldsymbol{\theta},c_m}(\mathbf{x}_m)} \frac{\partial f_{\boldsymbol{\theta},c_m}(\mathbf{x}_m)}{\partial \boldsymbol{\theta}} \right]$$

where $c_m$ denotes the class of $\mathbf{x}_m$, e.g. if $\mathbf{x}_m$ belongs to the third class, then $c_m = 3$. $f_{\boldsymbol{\theta},c_m}(\mathbf{x}_m)$ denotes the $c_m^{th}$ dimensional element of $f_{\boldsymbol{\theta}}(\mathbf{x}_m)$. We want a lower bound of $f_{\boldsymbol{\theta},c}(\mathbf{x})$ exists, so that the gradient $\nabla_{\boldsymbol{\theta}} \mathcal{L}_m(f_{\boldsymbol{\theta}})$ can be further bounded.

Then we can prove similar theorems just follow the steps in regression case.

## G    PROOF OF THEOREM 5

**Theorem 5** *Let $f_{\boldsymbol{\theta}}$ be a neural network with $L$ hidden layers, with each layer being either fully-connected or convolutional. Assume that $\|\mathcal{L}\|_\infty < \infty$. Then, $error(T) = |\widetilde{\mathcal{E}}(T, f_{\boldsymbol{\theta}}) - \overline{\mathcal{E}}(T, f_{\boldsymbol{\theta}})|$ is a non-decreasing function of $T$. Furthermore, for arbitrary $T > 0$ we have:*

$$error(T) \leq O\big(T^{2L+3}\big).$$

**Proof** Recall that $\overline{\mathcal{E}}(t, f_{\boldsymbol{\theta}})$ is defined based on $f_{m,\boldsymbol{\theta}}^t$, which is the resulting function whose parameters evolve according to the gradient flow $\dfrac{d\boldsymbol{\theta}_m^t}{dt} = -\nabla_{\boldsymbol{\theta}_m^t} \mathcal{L}(f_{m,\boldsymbol{\theta}}^t, \mathcal{D}_m^{tr})$.

We actually have the following (Santambrogio, 2016):

$$\|\triangle \boldsymbol{\theta}\| = \|\boldsymbol{\theta}^0 - \boldsymbol{\theta}^t\| \leq O(\sqrt{t}).$$

For simplicity and clearness, we use $\triangle$ to denote the change of any vectors and matrices. Thus, we know that

$$\|\triangle W^h\|_2 \leq \|\triangle W^h\|_{\mathcal{F}} \leq \|\triangle \boldsymbol{\theta}\| \leq O(\sqrt{t}).$$

Just like the proofs of Lemma 7, Lemma 8 and Lemma 9, we show that

$$\|\triangle g^h\| \leq O(t^{h/2}), \|\triangle b^h\| \leq O(t^{(L-h+1)/2}), \left\|\triangle \frac{\partial f_{\boldsymbol{\theta}}(\mathbf{x})}{\partial \boldsymbol{\theta}}\right\|_{\mathcal{F}} \leq O(t^{(L+1)/2}\sqrt{L+1})$$

by mathematical inductions; we skip the details here. Note that different from some previous theorem, here we focus on time t, and thus hide the effect of the spectral norms by treating them as constants.

Then, we have

$$\left\| \triangle\Big(\nabla_{\boldsymbol{\theta}}\mathcal{L}_m(f_{\boldsymbol{\theta}})\Big)\right\|$$

$$=\|\nabla_{\boldsymbol{\theta}^t}\mathcal{L}_m(f^t_{m,\boldsymbol{\theta}}) - \nabla_{\boldsymbol{\theta}}\mathcal{L}_m(f_{\boldsymbol{\theta}})\|$$

$$=\left\|\mathbb{E}_{(\mathbf{x}_m,\mathbf{y}_m)}\Big\{\left[f^t_{m,\boldsymbol{\theta}}(\mathbf{x}_m) - \mathbf{y}_m\right]\frac{\partial f^t_{m,\boldsymbol{\theta}}(\mathbf{x}_m)}{\partial\boldsymbol{\theta}^t}^{\mathsf{T}} - [f_{\boldsymbol{\theta}}(\mathbf{x}_m) - \mathbf{y}_m]\frac{\partial f_{\boldsymbol{\theta}}(\mathbf{x}_m)}{\partial\boldsymbol{\theta}}^{\mathsf{T}}\Big\}\right\|$$

$$=\left\|\mathbb{E}_{(\mathbf{x}_m,\mathbf{y}_m)}\Big\{\triangle f_{\boldsymbol{\theta}}(\mathbf{x}_m)\left[\frac{\partial f^t_{m,\boldsymbol{\theta}}(\mathbf{x}_m)(\mathbf{x}_m)}{\partial\boldsymbol{\theta}^t} + \triangle\frac{\partial f_{\boldsymbol{\theta}}(\mathbf{x}_m)}{\partial\boldsymbol{\theta}}\right]^{\mathsf{T}} + [f_{\boldsymbol{\theta}}(\mathbf{x}_m) - \mathbf{y}_m]\triangle\frac{\partial f_{\boldsymbol{\theta}}(\mathbf{x}_m)}{\partial\boldsymbol{\theta}}^{\mathsf{T}}\Big\}\right\|$$

$$\leq O(t^{L+1}\sqrt{L+1}).$$

Recall that:

$$\overline{\mathcal{E}}(T, f_{\boldsymbol{\theta}}) = \mathbb{E}_{\mathcal{T}_m}\left[\mathcal{L}_m(f^T_{m,\boldsymbol{\theta}})\right]$$

$$= \mathbb{E}_{\mathcal{T}_m}\left[\mathcal{L}_m(f_{\boldsymbol{\theta}}) + \int_0^T \nabla_t\mathcal{L}_m(f^t_{m,\boldsymbol{\theta}})\mathrm{d}t\right]$$

$$= \mathbb{E}_{\mathcal{T}_m}\left[\mathcal{L}_m(f_{\boldsymbol{\theta}}) + \int_0^T \frac{\mathrm{d}\boldsymbol{\theta}^t}{\mathrm{d}t}\nabla_{\boldsymbol{\theta}^t}\mathcal{L}_m(f^t_{m,\boldsymbol{\theta}})\mathrm{d}t\right]$$

$$= \mathbb{E}_{\mathcal{T}_m}\left[\mathcal{L}_m(f_{\boldsymbol{\theta}}) - \int_0^T \left\|\nabla_{\boldsymbol{\theta}^t}\mathcal{L}_m(f^t_{m,\boldsymbol{\theta}})\right\|^2\mathrm{d}t\right]$$

and

$$\widetilde{\mathcal{E}}(T, f_{\boldsymbol{\theta}}) = \mathbb{E}_{\mathcal{T}_m}\left[\mathcal{L}_m(f_{\boldsymbol{\theta}}) - T\|\nabla_{f_{\boldsymbol{\theta}}}\mathcal{L}_m(f_{\boldsymbol{\theta}})\|^2_{\mathcal{H}}\right] = \mathbb{E}_{\mathcal{T}_m}\left[\mathcal{L}_m(f_{\boldsymbol{\theta}}) - T\|\nabla_{\boldsymbol{\theta}}\mathcal{L}_m(f_{\boldsymbol{\theta}})\|^2\right].$$

Because

$$\widetilde{\mathcal{E}}(T, f_{\boldsymbol{\theta}}) - \overline{\mathcal{E}}(T, f_{\boldsymbol{\theta}})$$

$$= \int_0^T \left\|\nabla_{\boldsymbol{\theta}^t}\mathcal{L}_m(f^t_{m,\boldsymbol{\theta}})\right\|^2\mathrm{d}t - T\|\nabla_{\boldsymbol{\theta}}\mathcal{L}_m(f_{\boldsymbol{\theta}})\|^2$$

$$= \int_0^T \left\|\nabla_{\boldsymbol{\theta}}\mathcal{L}_m(f_{\boldsymbol{\theta}}) + \triangle\Big(\nabla_{\boldsymbol{\theta}}\mathcal{L}_m(f_{\boldsymbol{\theta}})\Big)\right\|^2\mathrm{d}t - T\|\nabla_{\boldsymbol{\theta}}\mathcal{L}_m(f_{\boldsymbol{\theta}})\|^2$$

$$= \int_0^T 2\nabla_{\boldsymbol{\theta}}\mathcal{L}_m(f_{\boldsymbol{\theta}})\triangle\Big(\nabla_{\boldsymbol{\theta}}\mathcal{L}_m(f_{\boldsymbol{\theta}})\Big)^{\mathsf{T}} + \left\|\triangle\Big(\nabla_{\boldsymbol{\theta}}\mathcal{L}_m(f_{\boldsymbol{\theta}})\Big)\right\|^2\mathrm{d}t,$$

we have

$$error(T) = |\widetilde{\mathcal{E}}(T, f_{\boldsymbol{\theta}}) - \overline{\mathcal{E}}(T, f_{\boldsymbol{\theta}})| \leq O\Big(\frac{L+1}{2L+3}T^{2L+3}\Big) = O\big(T^{2L+3}\big)$$

by simple calculation.

On the other hand, observe that

$$\bar{\mathcal{E}}(T, f_{\boldsymbol{\theta}}) = \mathbb{E}_{\mathcal{T}_m}\left[\mathcal{L}_m(f_{\boldsymbol{\theta}}) - \int_0^T \left\|\nabla_{\boldsymbol{\theta}^t}\mathcal{L}_m(f^t_{m,\boldsymbol{\theta}})\right\|^2_{\mathcal{H}}dt\right],$$

$$\tilde{\mathcal{E}}(T, f_{\boldsymbol{\theta}}) = \mathbb{E}_{\mathcal{T}_m}[\mathcal{L}_m(f_{\boldsymbol{\theta}}) - T\left\|\nabla_{\boldsymbol{\theta}}\mathcal{L}_m(f_{\boldsymbol{\theta}})\right\|^2_{\mathcal{H}}].$$

We let

$$G(\tau) = \int_0^{\tau} \left\|\nabla_{\boldsymbol{\theta}^t}\mathcal{L}_m(f^t_{m,\boldsymbol{\theta}})\right\|^2 dt,$$

and assume that $\nabla_{\boldsymbol{\theta}^t}\mathcal{L}_m(f^t_{m,\boldsymbol{\theta}})$ is continuous at $t = 0$. Then, we have $G'(\tau) = \|\nabla_{\boldsymbol{\theta}^t}\mathcal{L}_m(f^t_{\boldsymbol{\theta}})\|^2$.

$$\left\|\bar{\mathcal{E}}(T, f_{\boldsymbol{\theta}}) - \tilde{\mathcal{E}}(T, f_{\boldsymbol{\theta}})\right\| = \left\|\mathbb{E}_{\mathcal{T}_m}\left[\int_0^T \left\|\nabla_{\boldsymbol{\theta}^t}\mathcal{L}_m(f^t_{m,\boldsymbol{\theta}})\right\|^2 dt - T\left\|\nabla_{\boldsymbol{\theta}}\mathcal{L}_m(f_{\boldsymbol{\theta}})\right\|^2_{\mathcal{H}}\right]\right\|$$

$$= \left\|\mathbb{E}_{\mathcal{T}_m}(G(T) - T \cdot G'(0))\right\|,$$

where $TG'(0) = G(0) + TG'(0)$ (note that $G(0) = 0$) is a first order approximation to $G(T)$ at $\tau = 0$. When $T = 1$, $G(T) - TG'(0)$ can be taken as a local truncation error (i.e., the error that occurs in one step of a numerical approximation). When $T$ increases, the difference is no better than the global truncation error (in $T$ steps):

$$\left\| G(T) - \sum_{i=0}^{T}(i - (i-1))G'(i) \right\| = \left\| \sum_{i=0}^{T}\int_{i}^{i+1} \left\|\nabla_{\boldsymbol{\theta}^t}\mathcal{L}_m(f_{m,\boldsymbol{\theta}}^t)\right\|^2 - \left\|\nabla_{\boldsymbol{\theta}^i}\mathcal{L}_m(f_{m,\boldsymbol{\theta}}^{t=i})\right\|^2 dt \right\|$$

$$\approx \left\| \sum_{i=0}^{T}\int_{i}^{i+1} 2 \cdot \triangle_t^i\mathcal{L}_m(f_{m,\boldsymbol{\theta}}) \cdot \nabla\mathcal{L}_m(f_{m,\boldsymbol{\theta}}^t)dt \right\|,$$

where $\triangle_t^i\mathcal{L}_m(f_{m,\boldsymbol{\theta}}) = \nabla_{\boldsymbol{\theta}^t}\mathcal{L}_m(f_{m,\boldsymbol{\theta}}^t) - \nabla_{\boldsymbol{\theta}^i}\mathcal{L}_m(f_{m,\boldsymbol{\theta}}^t)$ as shown previously , $i$ is the $i$-th time step, and $G'(i)$ is the gradient of $G$ at time step $i$. Now we can see that $\left\|\overline{\mathcal{E}}(T, f_{\boldsymbol{\theta}}) - \tilde{\mathcal{E}}(T, f_{\boldsymbol{\theta}})\right\|$ highly relates to the difference between $\nabla_{\boldsymbol{\theta}^t}\mathcal{L}_m(f_{m,\boldsymbol{\theta}}^t)$ at different time steps (i.e. $\triangle_t^i\mathcal{L}_m(f_{m,\boldsymbol{\theta}})$), $\nabla_{\boldsymbol{\theta}^t}\mathcal{L}_m(f_{m,\boldsymbol{\theta}}^t)$ and $T$ . The first two terms relate to how flat or sharp the hyperplane of $\mathcal{L}_m(f_{m,\boldsymbol{\theta}})$ is near $t = 0$. We can wrap it as a constant $C_0(\mathcal{L}, t = 0)$. Then, the error is at least $C_0(\mathcal{L}, t = 0) \cdot O(T)$. For the hyperplane smooth enough, we can further get a first order approximation of $\triangle_t^i\mathcal{L}_m(f_{m,\boldsymbol{\theta}})$ and yield $C(\mathcal{L}, t = 0)O(T^2)$, where $C(\mathcal{L}, t = 0)$ can be analogized as the second order derivative of $\mathcal{L}$. $\blacksquare$

## H  SOME EXPERIMENTAL DETAILS

### H.1  IMPLEMENTATION OF CLASSIFICATION FOR META-RKHS-II

As we mentioned earlier, our proposed energy functional with closed form adaptation can not be directly applied to classification problem. We handle this challenge following Arora et al. (2019). For a $d_y$ class classification problem, every data $\mathbf{x}$ is associated with a $R^{d_y}$ one-hot vector $\mathbf{y}$ as its label. For $C$ classes classification problem, its encoding is $C$ dimensional vector and we use $-1/C$ and $(C - 1)/C$ as its correct and incorrect entries encoding. In the prediction, $Y^{tr}$ is replaced by the encoding of training data. $f_{\boldsymbol{\theta}}(\mathbf{x})$ is replaced by $f_{\boldsymbol{\theta}}(\mathbf{x})^{\intercal}[1, ..., 1] \in R^{n \times d_y}$ for dimension consistency. During the testing time, we compute the encoding of the test data point, and choose the position with largest value as its predicted class.

## I  EXTRA EXPERIMENTAL RESULTS

### I.1  COMPARISON WITH RBF KERNEL

One interesting question is, without introducing extra model components or networks, what will the results of other kernel be? We provide the results of using RBF (Gaussian) kernel here: $42.1 \pm 1.9$ (5-way 1-shot) and $54.9 \pm 1.1$ (5-way 5-shot) on Mini-ImageNet, $32.4 \pm 2.0$ (5-way 1-shot) and $38.2 \pm 0.9$ (5-way 5-shot) on FC-100, which are worse than the NTK based Meta-RKHS-II, showing the superiority of using NTK.

### I.2  MORE RESULTS ON OUT-OF-DISTRIBUTION GENERALIZATION

We provide some more results on out-of-distribution generalization experiments here. From the results we can find that the proposed methods is more robust and can generalize to different datasets better.

Table 5: Meta testing on different out-of-distribution datasets with model trained on FC-100.

| | 5 WAY 1 SHOT | | 5 WAY 5 SHOT | |
|---|---|---|---|---|
| ALGORITHM | CUB | VGG FLOWER | CUB | VGG FLOWER |
| MAML | $31.58 \pm 1.89\%$ | $50.82 \pm 1.94\%$ | $41.72 \pm 1.29\%$ | $65.19 \pm 1.36\%$ |
| FOMAML | $32.34 \pm 1.57\%$ | $49.90 \pm 1.78\%$ | $41.96 \pm 1.53\%$ | $66.87 \pm 1.45\%$ |
| REPTILE | $33.56 \pm 1.40\%$ | $46.77 \pm 1.81\%$ | $42.79 \pm 1.38\%$ | $67.97 \pm 0.71\%$ |
| IMAML | $32.49 \pm 1.52\%$ | $49.96 \pm 1.98\%$ | $38.92 \pm 1.62\%$ | $59.80 \pm 1.82\%$ |
| BAYESIAN TAML(SOTA) | $31.82 \pm 0.49\%$ | $49.58 \pm 0.55\%$ | $43.97 \pm 0.57\%$ | $67.36 \pm 0.53\%$ |
| META-RKHS-I | $34.12 \pm 1.34\%$ | $48.81 \pm 1.89\%$ | $43.31 \pm 1.43\%$ | $69.02 \pm 0.62\%$ |
| META-RKHS-II | $\mathbf{36.35 \pm 1.07}\%$ | $\mathbf{59.75 \pm 1.23}\%$ | $\mathbf{49.92 \pm 0.68}\%$ | $\mathbf{76.32 \pm 0.58}\%$ |

### I.3    More Results on Adversarial Attack

We now show some more extra results on adversarial attack in the following figures. Consistent to the results in main text, we can find that our proposed methods are more robust to adversarial attacks.

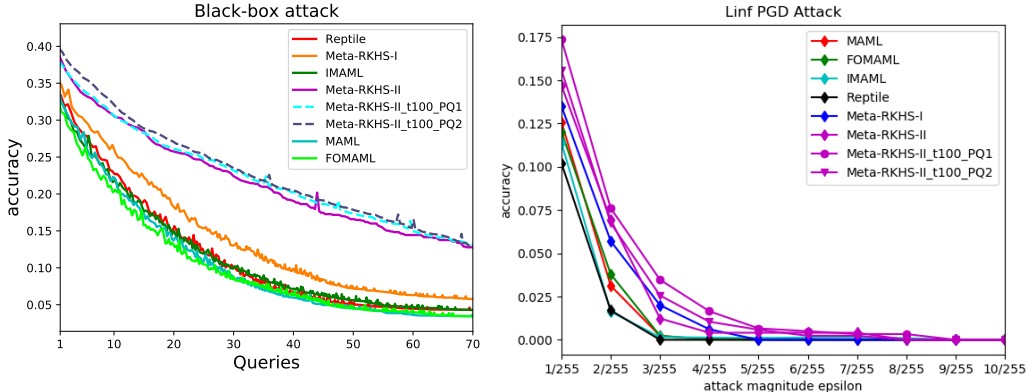

Figure 6: FC-100 5-way 5-shot Black-box attacks (left) and 5-way 1-shot PGD $\ell_\infty$ norm attack (right).

### I.4    Impact of Gradient Norm in Meta-RKHS-I

In this experiment, we compare between our proposed Meta-RKHS-I and Reptile. We evaluate the trained models with different adaptation steps in testing-time. The comparison is shown in Figure 7. As we can see, our Meta-RKHS-I always gets better results than Reptile, which supports our idea that the learned function should be close to task-specific optimal and have large functional gradient norm. These two conditions together lead to the ability of fast adaptation.

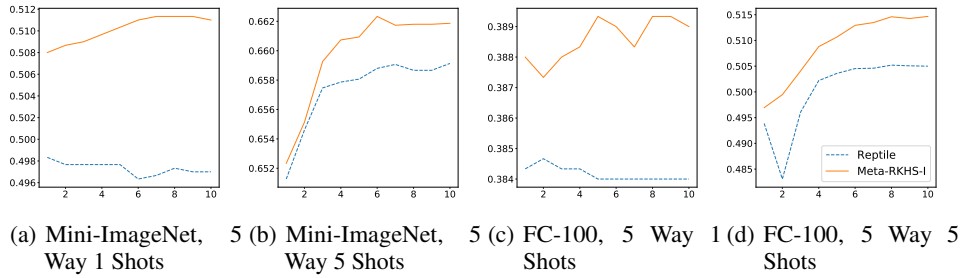

(a) Mini-ImageNet, 5 Way 1 Shots    (b) Mini-ImageNet, 5 Way 5 Shots    (c) FC-100, 5 Way 1 Shots    (d) FC-100, 5 Way 5 Shots

Figure 7: Reptile (dashed) vs. Meta-RKHS-I (solid) with different testing adaptation steps (x-axis).

### I.5    Impact of network architecture for different meta-learning models

In this section, we compare different meta-learning models with feature channels of 100 and 200 of the CNN network structure with 4 or 5 CNN layers respectively.

Table 6: Few-shot classification results on Mini-ImageNet with different number of feature channels of 4 convolution layers.

| ALGORITHM | 100 | | 200 | |
|---|---|---|---|---|
| | 5 WAY 1 SHOT | 5 WAY 5 SHOTS | 5 WAY 1 SHOT | 5 WAY 5 SHOTS |
| MAML | $49.50 \pm 1.58\%$ | $64.31 \pm 1.07\%$ | $48.91 \pm 1.69\%$ | $63.96 \pm 0.82\%$ |
| FOMAML | $48.69 \pm 1.62\%$ | $63.73 \pm 0.76\%$ | $48.55 \pm 1.86\%$ | $63.18 \pm 0.96\%$ |
| iMAML | $49.30 \pm 1.94\%$ | $62.89 \pm 0.95\%$ | $48.23 \pm 1.58\%$ | $62.25 \pm 0.83\%$ |
| REPTILE | $50.20 \pm 1.69\%$ | $64.12 \pm 0.92\%$ | $48.72 \pm 1.97\%$ | $63.67 \pm 0.79\%$ |
| META-RKHS-I | $51.23 \pm 1.79\%$ | $66.69 \pm 0.73\%$ | $\mathbf{51.54 \pm 1.64}\%$ | $\mathbf{65.92 \pm 0.92}\%$ |
| META-RKHS-II | $\mathbf{51.37 \pm 2.31}\%$ | $\mathbf{66.97 \pm 0.98}\%$ | $50.96 \pm 2.15\%$ | $65.21 \pm 0.87\%$ |

Table 7: Few-shot classification results on Mini-ImageNet with different number of feature channels of 5 convolution layers.

| ALGORITHM | 100 | | 200 | |
|---|---|---|---|---|
| | 5 WAY 1 SHOT | 5 WAY 5 SHOTS | 5 WAY 1 SHOT | 5 WAY 5 SHOTS |
| MAML | $49.87 \pm 1.65\%$ | $65.78 \pm 1.18\%$ | $48.62 \pm 1.82\%$ | $63.25 \pm 0.75\%$ |
| FOMAML | $48.93 \pm 1.71\%$ | $64.37 \pm 0.80\%$ | $48.27 \pm 1.74\%$ | $62.95 \pm 0.83\%$ |
| iMAML | $48.03 \pm 1.76\%$ | $62.15 \pm 0.83\%$ | $47.52 \pm 1.73\%$ | $61.77 \pm 0.89\%$ |
| REPTILE | $50.62 \pm 1.83\%$ | $64.53 \pm 0.97\%$ | $49.33 \pm 1.89\%$ | $63.26 \pm 0.70\%$ |
| META-RKHS-I | $\mathbf{52.45 \pm 1.88}\%$ | $66.07 \pm 0.69\%$ | $\mathbf{51.37 \pm 1.92}\%$ | $\mathbf{65.39 \pm 0.98}\%$ |
| META-RKHS-II | $50.92 \pm 2.16\%$ | $\mathbf{66.45 \pm 0.91}\%$ | $50.43 \pm 2.42\%$ | $64.17 \pm 1.06\%$ |

