# OpenReview forum: "Meta-Learning with Neural Tangent Kernels"
_ICLR.cc/2021/Conference — ICLR 2021 Poster_

### Official Review · AnonReviewer4 · 2020-10-23
**This is interesting paper that would benefit from revision as described in the review**

**Rating:** 7
**Confidence:** 5

**Review:**

The authors propose two meta-learning algorithms in the reproducing kernel Hilbert space (RKHS) induced by the recently proposed Neural Tangent Kernels (NTK). The authors show how their algorithms obviate an explicit inner loop or task-adaptation step in the meta-learning training phase. In first algorithm, no explicit adaptation function is used, whereas in the second, a close form adaptation function which invokes the NTK is proposed - which is a simpler adaptation than that of MAML and hence, offers computational efficiency. The work is interesting and  supported by theory inspired from the NTK theory, and adds to the newly expanding literature in the use of kernels in meta-learning (unlike the authors’ claim in the introduction, theirs is not the first meta-learning paradigm in the RKHS cf (Wang et al 2020, Cerviño et al 2019)). The authors perform extensive experiments on regression and classification datasets and compare their results with other MAML-type algorithms. The experimental results do not show significant gains in terms of performance over the existing MAML approaches, except in the case of out-of-distribution datasets and adversarial attacks, where it is shown to outperform the others. The performance similarity to other methods is not surprising since the proposed approaches can be seen as an efficient approximations of the MAML.

I give my detailed comments next:

-Firstly, I believe the title of the paper could be changed to ‘Meta-Learning in the RKHS induced by the NTK’ or something more specific.  Currently, it comes off as rather broad and disproportionate to the work and existing work.

-I had some issues with the claim that both the Meta-RKHS approaches do not need an explicit inner-loop adaptation: this does not seem true. In the case of MetaRKHS-I , an inner adaptation is not needed in the meta-training, but necessary just like the MAML during for a test task. In the case of MetaRKHS-II, the NTK gradient flow based adaptation in Eq(6) forms the inner-loop— just that it is a more efficient inner update than the MAML. The authors must consider rephrasing the claim to reflect these.

-	Meta-RKHS-I:
1)	From what I understand, the treatment is based on an approximation to the MAML with k-inner gradient descent steps through a Taylor expansion. To this extent, one can expect the performance to be similar to that of $k$-step MAML on using the step size $k\alpha$ in equation (4) - indeed we see this in the experimental results.

2)	It is unclear as to how once the meta-parameter $\theta$ is learnt, the parameters for a new task are obtained from it— from what I see, it is obtained by $k$-steps of actual gradient descent just like the MAML. This was not mentioned anywhere clearly in the manuscript, I estimated this from the experiment plots in Figure 5 of the Appendix which mentions different inner steps.

3) The Taylor expansion in of equation (2): Under what case is this expansion valid, does this again assume a high degree of similarity between the tasks?
4) Before equation (3): ‘…equation 2 is an unbiased estimator of …’ This claim is not immediately evident to me, would be better if the authors could expand a bit here
5) The connection to the NTK seems a bit weak and superficial — Based on eq (3), the authors propose the energy functional in eq(4) for learning the meta-parameter and this is where the RKHS comes in. However, in the very next paragraph in Theorem 2, the equivalence of the gradient in parameter space and the functional gradient is claimed. Here I do not see what properties of the NTK are being invoked. Eq (4) can be with the the parameter gradient instead of the gradient in the RKHS and the approach will continue to hold valid. The authors should bring out the connection to RKHS and NTK more clearly, at present, it seems to me that the approach does not explicitly have connections to an RKHS. Indeed, the proofs of Theorem 3 and 4 also do not seem to use the properties of RKHS or the NTK.

-Meta-RKHS-II
Here, the authors propose an adaptation function based on the NTK and the gradient flow. This is an interesting adaptation function that evokes the NTK and in the process can help approximate a $k$-order inner gradient and yet be free of the computational difficulties that come due to this in the standard MAML systems. Unlike Meta-RKHS-I, the connection to NTK is strong and clear here. The authors should consider expanding and emphasising this portion better. For example, what is the implication of Theorem 5?
-Last paragraph on page 5: ‘Intuitively,…thus can be deem more robust.’ I do not follow this, could the authors expand here? I also found the robustness discussion to be needing clarity on the whole.

-	Figure I:  The axes and the legends are not readable.

-Section 4.3 and 4.4: The Meta-RKHS methods significantly outperform other approaches in the case of adversarial attacks and out-of-the-distribution tasks. This is of merit since it is known that MAML type approaches are far too sensitive to the outlier tasks. Can this robustness be better explained or mathematically analysed in terms of the RKHS or the NTK? This would greatly strengthen the contribution.

Overall, I feel that this is an interesting  and novel contribution, particularly in terms of mathematical concepts, though the approach does not necessarily outperform similar methods by a significant margin except in the case of out-of-the-distribution tasks or adversarial attacks. The connection to RKHS seems a bit weak and currently appears in the form of NTK and that too evidently only in MetaRKHS-II. A suggestion is also consider dataset cases where MAML necessarily requires multiple inner gradient steps to just one step inner gradient update ( the first order MAML) . (Currently all the examples in the paper show similar performance for both MAML and first order MAML). This could help verify if the Meta-RKHS approaches can indeed achieve similar performance as multi-step inner gradient MAML, while having much lower complexity.


References:
(Wang et al 2020)  Haoxiang Wang, Ruoyu Sun, and Bo Li. Global convergence and induced kernels of gradient-based meta-learning with neural nets, 2020.

(Cerviño et al 2019) J. Cerviño, J. A. Bazerque, M. Calvo-Fullana and A. Ribeiro, "Meta-Learning through Coupled Optimization in Reproducing Kernel Hilbert Spaces," 2019 American Control Conference (ACC), Philadelphia, PA, USA, 2019, pp. 4840-4846, doi: 10.23919/ACC.2019.8814419.

---

> ### Author Response · Authors · 2020-11-20
> **Response to Reviewer 4**
>
> We thank this reviewer for the inspiring comments, which are summarized below, along with our responses, clarifications and additional experimental results.
>
> Major concerns and our responses:
>
> 1) About the title.
>
>     Response: Thanks for the suggestion. We have changed the title to “Meta Learning with Neural Tangent Kernels”.
>
> 2) About Meta-RKHS-I.
>
>     Response: Thanks for pointing out some potential issues. We address them below and have added corresponding clarifications in the revision.
>     1) The Meta-RKHS-I is inspired by the Talor expansion of MAML’s meta-objective. To connect it with NTK, we re-write the problem in the RKHS induced by NTK, and only consider the major component, i.e. the functional evaluated given meta-model and the gradient norm.
>     2) In meta-testing of Meta-RKHS-I, one needs to apply gradient descent on the testing tasks for adaptation, just like other gradient based methods.
>     3) The Taylor expansion approximation is valid when different order derivatives (e.g. gradient vector, Hessian matrix) of loss w.r.t the meta-parameter have relatively small norm (for all tasks), especially for the higher-order terms. Intuitively, it is  similar to assuming that the loss has a smooth landscape around the meta-parameter, on all the tasks. It is likely that when the tasks are similar to each other, we have a higher possibility to learn such a meta-parameter.
>     4) We mean that the RHS of (2) is an unbiased estimator of (3), because if one takes the expectation w.r.t the data samples ($\mathcal{D}_m^{tr}$ and $\mathcal{D}_m^{test}$), the expectation value equals (3), which actually comes from the independence of data samples.
>     5) Meta-RKHS-I is inspired from the Taylor expansion of MAML’s meta-objective, which only considers the major component in the RKHS. Meta-RKHS-I may not be as close to NTK as Meta-RKHS-II. But the equivalence between gradient norm in parameter space and functional gradient norm only holds in the RKHS *induced by the NTK*. It does not hold in arbitrary RKHS. As we revealed in Theorem 5, the two algorithms can be related to each other. Meta-RKHS-II actually contains an integral term of the functional gradient norm w.r.t time t (which can be found in the proof of Theorem 5. We have emphasized this in the revision).
>
> 3) About Meta-RKHS-II.
>
>     Response: Theorem 5 indicates that our two proposed algorithms can be related. The connection is closer for a smaller network (small L) and shorter adaption (small T), which coincides with our intuition and also is verified by the different performance in the experiments. Actually, the difference between the meta-objectives (energy functional) mainly comes from the fact that Meta-RKHS-I’s objective can be regarded as an approximation of the time-discrete adaptation, and Meta-RKHS-II is based on the time-continuous adaptation. The difference between the two objectives increases along time T. There is an upper bound related to T, which is related to the error of solving gradient flow by discrete-time scheme [1]. We have added clarification in the revision.
>
>
> [1]. Filippo Santambrogio.  Euclidean, Metric, and Wasserstein  gradient flows.

---

> > ### Author Response · Authors · 2020-11-20
> > **Response to Reviewer 4**
> >
> > 4) On robustness of Meta-RKHS-II.
> >
> >     Response: Currently we only focus on explaining the robustness of Meta-RKHS-II from an intuitive perspective, as current theories of robustness machine learning and NTK are insufficient to explain the phenomenon. A theoretical explanation would call for substantial extra effort and is thus out of the scope of this paper.
> >     Our argument of the model being more robust is based on an intuitive perspective, based on some properties of both meta learning and NTK:
> >     1) Strong initialization (meta model): For NTK to generalize well, we argue that it is necessary to initialize the model with a good initialization. This is automatically achieved in our meta learning setting because the meta model serves as the initialization for NTK prediction, which will be optimized during training. Actually, this has been supported by recent research [2], which shows that there is a chaotic stage in the NTK prediction with finite neural networks, and the NTK regime can start immediately with a good initialization. This finding is actually quite inspiring and fits our meta learning setting well. The finding of [2] was not published when we submitted our paper, thus we did not cite it. We have revised this section to include this reference in our new version.
> >     2) Low complex classification boundary: It is known that NTK is a linear model in the NTK regime. Intuitively, generating adversarial samples with a lower complex model should be relatively harder because there is less data in the vicinity of the decision boundary, making the probability of the model being attacked lower. Since our above discussion reveals that our model is likely in the NTK regime with a good meta initialization,  we thus argue that our model can be more robust than the standard meta learning models.
> >     3) Our NTK-based model is robust enough to adapt with different time steps, while these finite time steps can be more robust to adversarial attacks than that of the infinite-time limit partially due to the complexity of the back-propagating gradients.
> > To sum up, we argue that the robustness of our model is due to the collective effect of the above three points, with formal analysis left for interesting future work.
> >
> >
> > 5) About the figure and references.
> >
> >     Response: Thanks for mentioning this. We will revise the figures to make it clearer. We have also included the suggested references.
> >
> >
> > [2] Fort et al., Deep learning versus kernel learning: an empirical study of loss landscape geometry and the time evolution of the Neural Tangent Kernel

---

### Official Review · AnonReviewer1 · 2020-10-26
**Good paper with minor problems**

**Rating:** 7
**Confidence:** 2

**Review:**

### Summary
In the attempt to create an adaptation-free meta-learning method, authors construct an RKHS based on the NTK and explain how to do (gradient-based/MAML-style) meta-learning in this space (instead of parameter space).
They propose two energy functionals (first one based on maximizing the norm of the parameters and the second one making the adaptation in closed form, based on the NTK) that approximate the MAML's infeasible learning objective. Evaluation of the functionals doesn't require evaluating the function outside of "current" parameters $\theta$, thus alleviating the need for explicit adaptation, which is a cause of technical problems in other methods.
Authors show theoretical arguments confirming their method is closely related to MAML as well as the results of extensive experiments in few-shot classification, regression and out-of-distribution generalization tasks. Authors claim the result of training their (second) method is robust to adversarial examples.

### Good points
1. The benefits of the presented method are obvious: problems associated with a long adaptation loop is a common yet unsolved problem contemporary methods struggle with.
2. Experiments are extensive and their results give a convincing argument that the method works.
3. I find the method I intuitive, as the "high norm of the parameters" coincides with parts of parameter space, from which one can adapt the most within few gradient steps (ie. MAML will be naturally seeking these places too).

### Overclaims
1. Authors claim their method is a first "single-looped" meta-learning algorithm. It's not clear from the paper what is formally meant by that; why is iMAML/Reptile/WarpGrad (Flennerhag et al. 2019, https://arxiv.org/abs/1909.00025) not "single-looped"? In general, a comment on WarpGrad, which learns a good-for-adaptation geometry through a gradient preconditioner, seems warranted.
2. Authors claim their method is robust to adversarial attacks. They demonstrate it using just two attacks: a black- and a white-box one, which I don't consider extensive enough. There is a known bias of adversarial defences guarding only against particular attack methods and being susceptible to others (cf. Athalye et al. 2018: Obfuscated Gradients Give a False Sense of Security or Uesato et al. 2018: Adversarial Risk and the Dangers of Evaluating Against Weak Attacks).
Furthermore, it's not clear that the chosen baselines are the best meta-learning methods out there in terms of adversarial attacks to compare to, which makes the claim of "our methods are much more robust to adversarial attacks than existing approaches" groundless.
3. Results on classification/out-of-distribution generalization are ok-ish to support the claims of the paper, but for completeness I believe it would be preferable to note the SOTA performance on these tasks. Otherwise, a careless reader may be under the impression that the SOTA is held by the authors' methods.

### Personal biases
I believe short adaptation unrolls is a dead-end. As we get to meta-learn harder/wider task distributions, there won't be any place in the parameter space from which we could achieve good fine-tuning performance without an expressive adaptation procedure.
This is particularly clear in meta-RL, where without doing gradient steps, the data we will be obtaining won't fully describe the MDP we need to solve; one may imagine a task with multiple rooms where only after performing adaptation we will pass the correct door and observe the actual task.

### Small
In Sec. 4.4 Nilsback & Zisserman (2008) should be a \citep.

---

> ### Author Response · Authors · 2020-11-20
> **Response to Reviewer 1**
>
> We thank this reviewer for the helpful comments, which are summarized below, along with our responses, clarifications and extra experimental results.
>
> Major concerns and our responses:
>
> 1) About the single-looped.
>
>     Response: Thanks for the comments and sorry for the confusion. Originally, we use single/double loop to indicate whether an algorithm needs two computation loops (one contained in the other) for optimization. After careful consideration, we find that the boundary of single loop and double loop is not clear enough. For example, Reptile can be considered as both single loop and double loop in our previous definition. As a result, we decide to abandon the use of single-loop (except in some cases that do not cause confusion). Instead, we motivate our method by *a more efficient adaptation that does not rely on chain-based adaptation*. To compare the actual efficiency, we also provide the computational complexity in Table 1.
>
>
> 2) More attacks to verify the robustness.
>
>     Response: Thanks for the suggestion. We have conducted more experiments with other strong attacks such as the BPDA attack[1] and SPSA attack[2]. Compared to the baselines, our Meta-RKHS-II consistently achieves the best robustness. These extra experimental results are provided in Figure 3 and 4 in the revision.
>
>
> 3) SOTA on classification/out-of-distribution generalization.
>
>     Response: To our best knowledge,  Bayesian TAML [3] is the-state-of-the-art method on out-of-distribution meta learning prediction. We have trained and tuned the model and updated the SOTA results on CUB and VGG-Flower in Table 3 of the revision. Again, our model performs better.
>
>
> 4) About personal bias.
>
>     Response: We agree on your view of adaptation unrolling. We share your thinking that to adapt to more distinct tasks, long-range adaptation needs to be considered. This is exactly one of our motivations. Our Meta-RKHS-II can partially achieve this as we can control on how far the adaptation can take.
>
>
> [1] Anish Athalye et al., Obfuscated Gradients Give a False Sense of Security: Circumventing Defenses to Adversarial Examples
>
> [2] Jonathan Uesato et al. Adversarial Risk and the Dangers of Evaluating Against Weak Attacks
>
> [3] Learning to Balance: Bayesian Meta-Learning for Imbalanced and Out-of-distribution Tasks

---

### Official Review · AnonReviewer2 · 2020-10-28
**More explanations need**

**Rating:** 5
**Confidence:** 4

**Review:**

This paper mainly deals with the computational issues of Model Agnostic Meta-Learning (MAML). Specifically, it proposes two meta-learning algorithms where the hypothesis class (i.e. the mapping function set) is defined in RKHS induced by NTK. Extensive experimental studies on many tasks (i.e. regression, few-shot image classification, robustness to adversarial attack, and out-of-distribution generalization) illustrate its superiority compared with other baselines.


#############################################################################################
pros:
1. Overall, this paper is well-written and organized.
2. This work is based on recent solid theoretical results (i.e. NTK) from the deep learning theory.
3. The proposed methods have promising performance empirically.

#############################################################################################
cons:
1. For the proposed first algorithm (i.e. Meta-RKHS-1), what is the connection or difference between the introduced regularizer in Eq.(4) and some commonly used regularizers (e.g. $||f||_{\mathcal{h}}^2$) in RKHS. Furthermore, does the objective function in Eq.(4) could become negative? Please give more explanation.
2.  For the second algorithm (i.e. Meta-RKHS-2), the authors claim that it can have a closed-form adaptation function. In my opinion, this is mainly because the loss function is squared loss, just like kernel least squared regression. If the loss changes to other losses, such as cross-entropy loss, it cannot get a closed-form solution. If so, the algorithm also needs double-looped optimization. Please clarify it.
3. Although this paper mainly focuses on the NTK-induced RKHS, other kernel functions (e.g. RBF) also can hold for these two algorithms. It is interesting to test whether NTK is better than RBF in the meta-learning setting, such as the regression task.
4. The formal proofs for the theorems in this paper are mainly based on previous results. It is better to summarize the technical differences for clear theoretical contributions.

---

> ### Author Response · Authors · 2020-11-20
> **Response to Reviewer 2**
>
> We thank this reviewer for the helpful comments. Please see our responses below and some additional experiments in the revision, which, we believe, can address the concerns.
>
> Major concerns and our responses:
>
> 1) About the regularizer and objective function.
>
>     Response: $\Vert f\Vert ^2_{\mathcal{H}}$ is often used to regularize the function and to prevent overfitting. It adds a trade-off between fitting and the complexity of the function. Our regularizer $-\Vert \nabla_f \mathcal{L} \Vert ^2_{\mathcal{H}}$, which is the functional gradient norm, focuses on fast decreasing of the loss functional, and thus is different from those that are for a different purpose. The objective function can be negative, but will not affect the optimization process.
>
>
> 2) Closed form solution of Meta-RKHS-II.
>
>     Response: Thanks for the comment. Yes, you are right. The adaptation has a closed form only in the case of MSE loss. This is also a common setting in existing work related to RKHS. Based on related works such as [1][2], we wish to argue that this does not cause issues in practice, due to the following reasons:
>     1) The goals of other loss can usually be implemented and achieved by MSE loss: For example, the authors in [1] use the MSE loss to implement the classification problem originally defined with the cross entropy loss.  They show no loss in the performance.
>     2) If one insists on other losses, the prediction can be easily solved by relying on the well-developed ODE solver: For example, the authors in [2] use the ODE solver to solve the function evolution with the cross entropy loss, which does not introduce additional complexity. If one further requires a single-looped solution, we can approximate the loss such that it can be solved in closed-form, e.g., using taylor expansion for approximation.
>     In our paper, we do not consider the approximate solution from the above second bullet. Instead, we focus on using the MSE loss to implement other losses such that a closed-form adaptation can be achieved. We have made this clear in our revision.
>
>
> 3) RBF kernel functions in meta learning.
>
>     Response: For a fair comparison, we provide the experimental results of using RBF kernels under the same setting (model size,
> network architecture etc). The results of RBF kernel is $42.1\pm 1.9$ (1 shot) and $54.9 \pm 1.1$ (5 shots) on Mini-ImageNet, $32.4 \pm 2.0$ (1 shot) and $38.2 \pm 0.9$ (5 shots) on FC-100, which are worse than that of NTK. Due to space limit, we have incorporated these results in the Appendix of the revision.
>
>
> 4) About the proof of the theorems.
>
>     Response: Thanks for the suggestion. The proof techniques used in our paper are based on some previous works such as  [1,3], but with major differences, which are summarized below:
>     1) Previous works typically assume that the neural network is Gaussian initialized, while we do not have such an assumption as we are trying to learn a good meta-initialization in the meta-learning setting.
>     2) Previous works try to investigate the behavior of models during training, while we focus on revealing the connection between different meta-learning algorithms.
>     3) Previous work focuses on single-task regression/classification problems, while we focus on the meta-learning problem.
> Due to such differences, our proof needs significant adaptations from the existing ones. We have emphasized this in the appendix of our revision.
>
> [1] Arora et al., On Exact Computation with an Infinitely Wide Neural Net.
>
> [2] Lee et al., Wide Neural Networks of Any Depth Evolve as Linear Models Under Gradient Descent.
>
> [3] Allen-Zhu et al., A Convergence Theory for Deep Learning via Over-Parameterization.

---

### Official Review · AnonReviewer3 · 2020-10-29
**A solid contribution**

**Rating:** 7
**Confidence:** 4

**Review:**

Summary

In this paper, the authors view MAML from the lens of Reproducing Hilbert Kernel Hilbert Spaces (RKHS) by applying tools from the theory of Neural Tangent Kernels (NTKs). Based on these insights, they develop two meta-learning algorithms that avoid gradient-based inner-loop adaptation. Their algorithms are theoretically grounded and exhibit improved empirical performance.

Overall, this is a solid contribution that should be of interest to the community. Thus, I recommend acceptance.

Strengths

This paper is generally well written and proceeds to develop insights into gradient-based few-shot adaptation on first principles from NTK theory. In particular, they establish that parameter adaptation trajectories are equivalent to functional trajectories in some RKHS under the induced NTK, which allows us to bring to bear tools and analysis from that field.

In particular, the authors provide rigorous mathematical derivations and show that gradient-based few-shot adaptation of the initialisation can be approximated without inner-loop adaptation under certain assumptions on the NTK (that it induces a relatively linear adaptation space), from which they derive two algorithms that avoid gradient-based inner-loop adaptation. They demonstrate empirically that the proposed algorithms perform better than similar algorithms on the standard few-shot learning setup on miniImagenet, as well as that the method is significantly more robust to adversarial robustness and enjoys substantially generalisation to out-of-distribution task.

I am particularly impressed by the second algorithm, Meta-RKHS-II, which derives a closed form-solution to gradient-based adaptation in RKHS that they then map back into parameter space via NTK. Meta-learning the closed-form solver can be thought of as learning a functional inference process for task adaptation. This provides a fresh new perspective and opens up for several new research directions; in particular, the authors mention that inference over the NTK can render the search for task models linear, which provides benefits both in terms of robustness and generalisation. They offer some empirical evidence that this is indeed the case.

Weaknesses

While I welcome the principled approach taken in this paper, it is somewhat underwhelming that the first algorithm (which avoids an n^3 complexity in the data) is derived from a first-order Taylor series expansion of the original MAML in parameter space. After all, Meta-RKHS-I can be motivated a first-order Taylor approximation to MAML without the need to involve RKHS. With that said, I do appreciate the effort taken to establish that this is equivalent to optimisation in function space. The gist of this algorithm is to convert MAML into a multi-task objective with a "regulariser" that tries to maximise the gradient norm at initialisation. While I believe the precise regulariser introduced here differs from other works, I am missing a discussion of similar works that also propose ways of regularising MAML updates [e.g. 1, 2, 3 and follow-ups]. Finally, I'm a bit unsure as to how Meta-RKHS-I behaves during meta-testing, given that it removes the adaptation loop at meta-training - does it simply use the meta-learner initialisation without adaptation or do you perform gradient descent on meta-test tasks?


Minor comments

- At times, the writing can be a bit aggressive. In the abstract, you claim 'superiority of your paradigm': improved performance on miniImagenet is not sufficient evidence to make such a strong claim.
- if meta-RKHS-I does not do any adaptation on meta-test tasks, I do not think it is fair to say that it replaces the inner-adaptation process: it removes it in favor of a multi-task solution. This may not be a good strategy beyond few-shot learning.
- You mention that NTKs can yield loss functionals that are convex, which seems appealing. I would suggest making this connection to your proposed algorithms stronger.
- Eq. 2: is the transpose on the wrong gradient vector? On the line above it, you use \nabla as a column vector.
- Eq. 4: in the paragraph following it, you speak of 'solving' Eq 4 - but Eq 4 is a functional and not a problem: do you mean min Eq. 4?
- Thm 2 is somewhat cryptic. What you mean to say is that the regulariser can be mapped into parameter space, so that E=M, correct?
- Thms 3 and 4 could use some discussion. What is the take-away? How likely are these conditions to hold?
- The coloured line in the robustness graphs are very hard to read - I initially thought that MAML achieved optimal robustness...

+++ Post-rebuttal +++
I've read the rebuttal and am content with the response. I'm maintain my recommendation to accept this paper.

References
[1]  Guiroy et. al.. Towards Understanding Generalization in Gradient-Based Meta-Learning. 2019.
[2] Khodak et. al. Provable Guarantees for Gradient-Based Meta-Learning. 2019.
[3] Zhou et. al. Efficient Meta Learning via Minibatch Proximal Update. 2019.

---

> ### Author Response · Authors · 2020-11-20
> **Response to Reviewer 3**
>
> We thank this reviewer for the  insightful comments, which are summarized below, along with our responses and clarifications.
>
> Major concerns and our responses:
>
> 1. Motivation of Meta-RKHS-I (can be directly obtained by Taylor expansion of MAML), and other works.
>
>     Response: Thanks for the comment. We agree that the idea of Meta-RKHS-I can be derived from the Taylor expansion of the MAML objective function and solved in the parameter space. In this paper, we aim to solve the meta-learning problem in RKHS induced by NTK. Thus, we re-formulate it in the function space and reveal its connection with related algorithms (MAML, Meta-RKHS-II). We have added a discussion with other related works in the revision.
>
>
> 2. How to perform meta-testing in Meta-RKHS-I ?
>
>     Response: Sorry for the confusion. We do need adaptations for Meta-RKHS-I in testing, which is done by gradient descent on meta-test tasks. We have clarified this in the revision.
>
>
> 3. Sometimes writing can be a bit aggressive.
>
>     Response: Thanks for the comments. We have made corresponding changes. For example, we have changed “superiority of our paradigm” to “advantage of our paradigm”.
>
>
> 4. NTK can yield loss functionals that are convex.
>
>     Response: Thanks for the suggestion. We have emphasized this in the revision.
>
>
> 5. Notation in Eq.2.
>
>     Response: For ease of  our later notation in the formulation and proofs, we write the gradient $\nabla_{\theta_i} \mathcal{L}$ (thus the parameter as well) as a row vector. We have added clarification in the revision.
>
>
> 6. About Eq.4.
>
>     Response: Thanks for pointing out this. Yes, we meant minimizing the functional, and have corrected it in the revision.
>
>
> 7. Thm 2 is somewhat cryptic. What you mean to say is that the regularizer can be mapped into parameter space, so that E=M, correct?
>
>     Response: Yes, you are right. Due to the property of NTK, the functional gradient norm in the RKHS induced by NTK is equivalent to the norm in the parameter space.
>
>
> 8. Take-away and conditions of Theorem 3 and 4.
>
>     Response: Theorem 3 and Theorem 4 indicate that for a meta-model with only fully-connected layers or convolutional layers, the proposed Meta-RKHS-I is a good approximation of MAML with bounded error.
>
>     For any neural network with arbitrary parameters, the theorems hold. This is because $\alpha < O(qr)$ can always be satisfied with some unknown constant hidden in the big-O notation. However, for a given $\alpha$, the final difference $O(1/L)$ might be large in certain cases, e.g. when the parameter matrices have very large spectral norms all the time. We have added some clarification in the revision.
>
>
> 9. The coloured line in the robustness graphs are very hard to read.
>
>     Response: Thanks for pointing this out. We have revised the figures to make it clearer.

---

### Author Response · Authors · 2020-11-20
**Author Response Summary**

We would like to thank all the reviewers for their thoughtful and inspiring comments. According to the reviews, we have done a major revision to address the reviewers’ concerns. All the changes have been incorporated into the manuscript and a new version of the paper has been uploaded to the openreview system. The changes are marked with “blue” in this revision for better illustration.  Below is a summary of the major changes:

1) We add more detailed discussions, especially on the robustness of Meta-RKHS-II.

2) We include more experimental results in the main text and appendix. Additional results with different adversarial attack methods including BPDA and SPSA are provided. More results are provided on out-of-distribution generalization, including comparisons with the current state-of-the-art model. Finally, an extra experiment with RBF kernel, instead of NTK, is conducted for meta learning.  All experiments still suggest that our models outperform others.

3) We add more details and clarifications, including how to perform meta-testing adaptation for Meta-RKHS-I, changes on  the confused use of “single-looped” and “double-looped”, and the connection between Meta-RKHS-I and Meta-RKHS-II.

---

### Decision · Program_Chairs · 2021-01-07
**Final Decision**

**Decision:**

Accept (Poster)

**Comment:**

This paper considers meta-learning based on MAML.  The authors use Neural Tangent Kernels (NTKs) to develop two meta-learning algorithms that avoid the inner-loop adaptation, which makes MAML computationally intensive.  Experimental results demonstrate favorable empirical performance over existing methods.

The paper is generally well written and readable.  The proposed methods are well motivated and based on solid theoretical ground.  The emprirical performance shows advantages in efficiency and quality.   This work is worth acceptence in ICLR 2021.